# Bridging the Data Provenance Gap Across Text, Speech, and Video

**Shayne Longpre, Nikhil Singh, Manuel Cherep, Kushagra Tiwary, Joanna Materzynska, William Brannon, Robert Mahari, Naana Obeng-Marnu, Manan Dey, Mohammed Hamdy, Nayan Saxena, Ahmad Mustafa Anis, Emad A. Alghamdi, Vu Minh Chien, Da Yin, Kun Qian, Yizhi Li, Minnie Liang, An Dinh, Shrestha Mohanty, Deividas Mataciunas, Tobin South, Jianguo Zhang, Ariel N. Lee, Campbell S. Lund, Christopher Klamm, Damien Sileo, Diganta Misra, Enrico Shippole, Kevin Klyman, Lester JV Miranda, Niklas Muennighoff, Seonghyeon Ye, Seungone Kim, Vipul Gupta, Vivek Sharma, Xuhui Zhou, Caiming Xiong, Luis Villa, Stella Biderman, Alex Pentland, Sara Hooker, Jad Kabbara**

The Data Provenance Initiative

## Abstract

Progress in AI is driven largely by the scale and quality of training data. Despite this, there is a deficit of empirical analysis examining the attributes of well-established datasets beyond text. In this work we conduct the largest and first-of-its-kind longitudinal audit across modalities—popular text, speech, and video datasets—from their detailed sourcing trends and use restrictions to their geographical and linguistic representation. Our manual analysis covers nearly 4000 public datasets between 1990-2024, spanning 608 languages, 798 sources, 659 organizations, and 67 countries. We find that multimodal machine learning applications have overwhelmingly turned to web-crawled, synthetic, and social media platforms, such as YouTube, for their training sets, eclipsing all other sources since 2019. Secondly, tracing the chain of dataset derivations we find that while less than 33% of datasets are restrictively licensed, over 80% of the source content in widely-used text, speech, and video datasets, carry non-commercial restrictions. Finally, counter to the rising number of languages and geographies represented in public AI training datasets, our audit demonstrates measures of *relative* geographical and multilingual representation have failed to significantly improve their coverage since 2013. We believe the breadth of our audit enables us to empirically examine trends in data sourcing, restrictions, and Western-centricity at an ecosystem-level, and that visibility into these questions are essential to progress in responsible AI. As a contribution to ongoing improvements in dataset transparency and responsible use, we release our entire multimodal audit, allowing practitioners to trace data provenance across text, speech, and video.

## 1 Introduction

The capabilities and flaws of multimodal foundation models are often directly attributable to their training data (Carlini et al., 2023a; Rando et al., 2022; Carlini et al., 2023b; Parmar et al., 2024; Liu et al., 2023b;a; Dai et al., 2024). While the importance of *data measurement* has been widely established by prior work (Gadre et al., 2024), so has a prevailing absence of data documentation (Gebru et al., 2021; Bender & Friedman, 2018), transparency (Bommasani et al., 2023), and detailed understanding (Dodge et al., 2021; Bandy & Vincent, 2021; Sambasivan et al., 2021)—especially for modalities other than text. A lack of thorough data analysis has led to significant challenges, including privacy issues (Subramani et al., 2023), retracting datasets with harmful content (Birhane et al., 2021; David, 2023), adversarially bypassing safety filters (Rando et al., 2022), facial recognition bias with respect to gender and skin type (Buolamwini & Gebru, 2018a), gender bias in hiring (Chang, 2023), benchmark contamination from overlapping train and test sets (Lee et al., 2023a), and challenges in copyright (Henderson et al., 2023). Understanding data provenance can aid mitigation attempts to

| | DATASETS | | SOURCES | | CREATOR ORGS | | LANGUAGES | | TASKS | LICENSES |
|---|---|---|---|---|---|---|---|---|---|---|
| | # | SIZE | # | DOMAINS | # | COUNTRIES | # | FAMILIES | | |
| TEXT | 3717 | 2.1T | 713 | 23 | 534 | 60 | 502 | 21 | 395 | 50 |
| SPEECH | 95 | 775k | 51 | 16 | 124 | 29 | 260 | 36 | 18 | 19 |
| VIDEO | 104 | 1.13M | 44 | 24 | 101 | 23 | - | - | 33 | 11 |
| TOTAL | 3916 | - | 798 | 83 | 659 | 67 | 608 | 37 | 443 | 55 |

Table 1: We quantify the breadth of our audit, including the total number of datasets (#), their size in tokens or hours, the sources, domains, creator organizations, countries, languages, tasks, and licenses. **In aggregate, we audited 3916 datasets from 659 organizations in 67 countries, spanning 2.1T tokens, and 1.9M hours. We cataloged nearly 798 unique sources, 443 tasks, and 55 licenses.**

reduce model bias and toxicity (Welbl et al., 2021; Pozzobon et al., 2023) address representation in data (Xu et al., 2021), contamination (Elazar et al., 2023), and quality (Kreutzer et al., 2022; Marion et al., 2023), as well as practical challenges with identifying copyright-free and permissively licensed sets (Min et al., 2023).

Despite the urgent need for the provenance and characteristics of widely used datasets, the majority of attention to date has centered on text datasets (Elazar et al., 2023; Longpre et al., 2024b), or a single feature such as prevalence of hate content (Dodge et al., 2021; Birhane et al., 2021). In contrast, in this work, we will critically examine several provenance features of data *across* text, speech, and video. We conduct the largest and most comprehensive multimodal audit of AI data, to date, reviewing nearly 4000 datasets between 1990-2024, covering 443 unique tasks, 608 languages, derived from 798 original sources, and constructed by 659 organizations, spanning 67 countries, over 1T tokens of text, and 1.9M hours of speech and video content (see Table 1).

There is an unprecedented acceleration in the development of multimodal AI systems, making all the more urgent an understanding of the datasets that underpin these breakthroughs. Our extensive collection of features from unstructured academic papers, websites, and repositories enables us to provide empirical grounding to an ambitious set of research questions surrounding data sourcing trends, intended licenses, and geographical and linguistic representation. Our key findings include:

1. **Multimodal data is increasingly sourced from the web, social media platforms, or synthetically generated;** rather than more curated sources such as movies, audiobooks or manually collected. These sources comprise the vast majority of text tokens, as well as speech and video hours in public data. However, while social media platforms provide data scale, heterogeneity and freshness by nature, they are also particularly prone to anti-crawling, copyright, privacy, and factuality concerns.
2. **Whereas only 25% of text, speech, and video datasets have non-commercial licenses, over 80% of content from each modality carries undocumented restrictions in the dataset's sources.** Dataset licenses are inconsistent with their source's restrictions for over 55% of content. Our audit provides the tools for multimodal developers to identify dataset restrictions, and apply their own standards.
3. **Geographical and linguistic representation have not improved for a decade, across the data ecosystem.** While the amount of data from under-represented creators and languages increases each year, to over 600 languages and 60 countries in 2024, their *relative representation* remains consistently western-centric, with no significant improvements from $> 0.7$ Gini coefficients. While Africa and South America organizations account for $< 0.2\%$ of all modality content, North America or European organizations span 93% of text tokens and 60%+ hours of speech and video.

Our work provides critical insights into the landscape of available multimodal data. We release the entire audit, collected data, and analysis tools, which we believe will bring immense value for data creators, developers, and researchers interested in promoting the responsible development of AI systems and analysis of the AI data ecosystem.

## 2 METHODOLOGY

While many prior works have surveyed the dataset ecosystem (Albalak et al., 2024; Liu et al., 2024c; Malik et al., 2021; Prabhavalkar et al., 2023; Li et al., 2019b), few empirically examine data corpora

at scale, and those that do focus present a more narrow focus around a specific feature like geographic bias or hate content(Birhane et al., 2023; McMillan-Major et al., 2022a; Shankar et al., 2017) or a single modality (Dodge et al., 2021; Caswell et al., 2021; Elazar et al., 2023; Longpre et al., 2024b). The goal of this work is to provide an empirical, ecosystem-level, and multimodal analysis of widely used training datasets (Cen et al., 2023). Our audit focuses on text, speech, and video, as prominent data modalities behind modern multimodal systems, such as Sora, Whisper, Gemini, GPT-4o, and others (Brooks et al., 2024; Zheng et al., 2024b; Radford et al., 2023; Peng et al., 2023; Team et al., 2023; OpenAI, 2024). Since training data for modalities can often be independent, multimodal models tend to interleave training batches with different combinations of one or two modalities (Aghajanyan et al., 2023). As such, we focus our analysis on datasets that represent one or a pair of these modalities.

**Annotation Features & Methodology**    In particular, we analyze data trends for the state of data permissions (licenses and terms), sourcing (the web, human annotation, and synthetic generation), and representation (of tasks, organizations, languages, and countries). We adopt Longpre et al. (2024b)'s methodology, including the license annotation taxonomy and process, to manually audit these features precisely and rigorously. We go beyond prior work, which considers dataset licenses, by extending the taxonomy to consider the terms of use of the sources of the dataset, either from models used to generate synthetic data (e.g. OpenAI's non-compete clause[1] or Meta's acceptable use policy for Llama 3.1[2]), or the source's policy on content restrictions, which can be conveyed in the form of a license, terms of use, or content policy on a website (Klyman, 2024). For each dataset, the source terms are annotated as Unrestricted, Unspecified, Source Closed or Model Closed, as defined in Table 2. For Figure 2 we combine Source Closed and Model Closed into *Restricted*.

As with prior work (Longpre et al., 2024b;c), we engage domain experts for these annotation tasks— AI researchers whose work pertains to the modality and topic. Because many datasets are iteratively re-packaged before they appear in their final form and often shared on popular dataset marketplaces like HuggingFace, Papers with Code or Github, prior work has found that relevant licensing terms or sourcing information for AI training data is frequently omitted (Longpre et al., 2024b). To ensure we collect this information, we require a full trace of metadata back to their original sources (sometimes a chain of github repositories, websites, or academic papers). This search can be onerous, especially for terms and licenses, but ensures rigor in the results. Table 1 enumerates the full statistics of our audit. All annotations and analysis code will be made publicly available on release.

**Scope & Dataset Selection**    For each modality, we define the scope of the audit (detailed separately below), then aggregate resources to distill a list of relevant datasets. The scope is focused on (a) publicly available datasets, (b) widely used tasks in the context of general-purpose model development, and (c) relevance to generative tasks. However, we do consider classification-based datasets in text, speech, and video that can and are frequently re-purposed for generative uses (e.g. instruction tuning). Within the defined audit scope, we use a mix of the HuggingFace Datasets platform, survey papers, survey repositories, workshop proceedings, and expert review to accumulate relevant datasets. More detail about the dataset selection and collection process is given for each modality below. Each modality requires its own independent process, by virtue of their community dataset ecosystems being unique (discussed in Section 4). Note that text has a wider heterogeneity of published publicly available datasets than speech or video. Typically those datasets have been aggregated into large, standardized text-to-text collections, and as such we trace both these *Text (Collections)* and their constituent *Text (Datasets)*. All datasets are described, linked, and attributed in Appendix D.

## 2.1   TEXT

**Scope**    We focus on providing an extensive audit for *post-training* datasets, used in training language models. We include single and multi-turn formats, encompassing both datasets typically used for instruction finetuning (SFT) and preference alignment Rafailov et al. (2023). This scope reflects the prominent role of general-purpose language models, which benefit from multi-task training on heterogeneous collections that span a variety of linguistic, reasoning, and knowledge intensive tasks like question answering, coding, tool use, translation, and classification (Wei et al., 2021; Ouyang et al., 2022).

---

[1]OpenAI Terms of Use
[2]Llama 3.1 Acceptable Use Policy

**Dataset Selection**  We expand the study conducted by the Data Provenance Collection (Longpre et al., 2024b), from 44 dataset collections (of 1858 supervised text datasets) to a superset of 108 collections of 3717 datasets, prioritizing recent, popular publicly available HuggingFace Datasets introduced between 2022 and April 2024. Our collection sourced popular datasets from recent survey papers (Albalak et al., 2024; Liu et al., 2024c) and tools (Longpre et al., 2024a). We additionally reviewed HuggingFace Datasets' most downloaded datasets every month, from April to July 2024, under the Natural Language Processing category, as well as the SFT/DPO datasets associated with popular open model releases. We also drew from major multilingual data repositories, including the SEACrowd Catalogue (Lovenia et al., 2024), the Masader Arabic Data Catalogue (Alyafeai et al., 2022), AI4Bharat (Kunchukuttan et al., 2020), and the Aya Collection (Singh et al., 2024a). Lastly, our list of datasets was reviewed and supplemented by language model experts to fill in notable omissions. In total, we trace the provenance and features of 3713 text datasets from 108 collections, covering 395 popular tasks, spanning from 1994 to 2024.

## 2.2 SPEECH

**Scope**  We audit speech datasets for which automatic speech recognition (ASR) was noted as a primary task. We focus on ASR datasets because: (1) ASR is fundamental to many speech technologies, including dictation tools, voice assistants, and chatbots (Aksënova et al., 2021; Zhang et al., 2022c); (2) large-scale speech datasets are typically designed for ASR (Li et al., 2023b); (3) ASR data follows standardized formats, making comparisons easier (e.g., corpus of audio clips paired with text); and (4) ASR data can often be reused for other tasks like text to speech (TTS) (Ito & Johnson, 2017) or language identification (Ardila et al., 2020b).

**Dataset Selection**  To curate a representative sample of popular ASR datasets, we relied on a combination of survey repositories[3], and HuggingFace Datasets using the "Automatic Speech Recognition" and "Text-to-Speech" task tags. We expanded coverage to well-documented datasets on the OpenSLR[4] platform, even if they were newer or less widely used. We expect this might reflect datasets that could be adopted more widely in the future. Finally, we included datasets related to low-resource languages and other languages not well-covered by our initial searches. Speech recognition models are increasingly highly multilingual Babu et al. (2021); Radford et al. (2023); Pratap et al. (2024), and datasets serving different communities of builders and end-users around the world are a priority for making speech recognition technologies more inclusive. In total, we trace the provenance and features of 95 speech datasets, covering 18 popular ASR tasks, spanning from 1990 to 2024.

## 2.3 VIDEO

**Scope**  Early video understanding models primarily focused on video classification, detection and action recognition, where short clips were categorized into predefined classes (Zheng et al., 2022; Zhu et al., 2020). More advanced tasks such as temporal action segmentation, video question answering, and video captioning were later introduced to build upon these foundational tasks (Moctezuma et al., 2022; Zhu et al., 2023). Recently, following the success in the field of image generation, video generation from text has become a new task that has shown promising results (Brooks et al., 2024; Zheng et al., 2024b; Blattmann et al., 2023; Esser et al., 2023). Given the scarcity of datasets for text-to-video and the often undocumented sources of data used in recent video generation models (Mauran, 2024), we take a broader approach to our collection of video datasets. We focus on annotating popular video tasks and limit our scope to datasets corresponding to video tasks that are either published, highly cited, or have 100+ downloads on HuggingFace. This approach is justified by three key factors: (1) the usefulness of video data to the research community stems from its collection and presentation in peer-reviewed work, (2) datasets can often be repurposed between different tasks, allowing for applicability to new tasks such as video generation from text, and (3) focusing on highly cited datasets ensures that datasets' quality and relevance has been validated by the research community.

**Dataset Selection**  We include datasets tagged with "Video Classification", "Text-to-Video", and "Video-Text-to-Text" from HuggingFace Datasets. We augmented this with datasets tagged by "Video Understanding" or "Video Generation" in PapersWithCode, as well as datasets listed in a popular Github survey repository. We also consulted the proceedings of recent video workshops: the Large Scale Video Understanding and Egocentric Vision workshops. We separately consulted a committee

---

[3]The Speech Datasets Collection

[4]openslr.org: Open Speech and Language Resources. OpenSLR is a widely used platform in the speech community, dedicated to hosting resources for speech tasks.

of non-author video experts to supplement the list with relevant datasets published at CVPR, ICCV, ECCV, and IJCV. In total, we trace the provenance and features of 104 video datasets, covering 33 popular video tasks, spanning from 2009 to 2024.

## 3 RESULTS

We discuss three key results related to (1) the rising use of web, social media and synthetic sources, (2) inconsistent and opaque restrictions on data use, and (3) a lack of improvement in geographical or linguistic representation. Each of these findings holds across modalities, at the ecosystem level.

### 3.1 RISING USE OF WEB, SOCIAL MEDIA & SYNTHETIC DATA

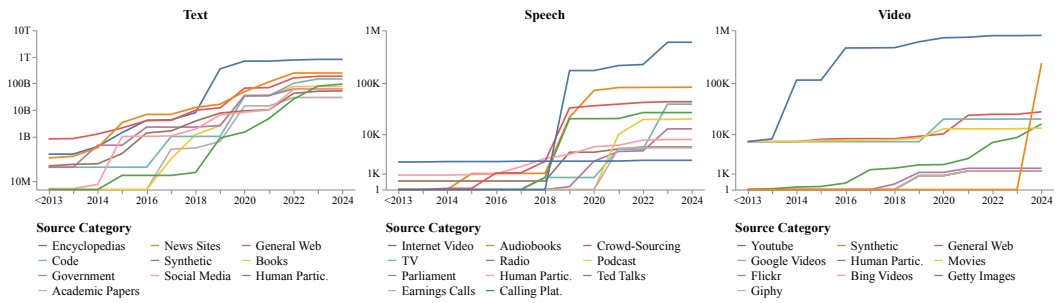

Figure 1: The cumulative size of data (log-scale tokens for text, hours for speech/video) from each source category, across modalities. The source categories in the legend are sorted descending by quantity. **Speech and video sources are increasingly dominated by internet videos and YouTube. Though text is especially web or encyclopedia (wiki) sources, synthetic text is rising in popularity.**

**The need for scale, and heterogeneity have driven rising use of data from web-crawled, social media, and synthetic data sources.** Developers have sought out ever larger and conveniently accessible sources of training data (Hoffmann et al., 2022; Henighan et al., 2020). While small, human-curated datasets are often sufficient and sometimes preferred due to higher quality, these sources often do not scale to present demands (Kaplan et al., 2020; Henighan et al., 2020). In Figure 1, we empirically measure the rising use of web crawling and social media (or "forum") websites that provide some of the most scalable and fresh content. While web-sourced data was always prominent, the balance of sources becomes much more skewed after 2018—note the use of the y-axis log scale. We find for Speech and Video that by far the most prominent source of data has become internet videos, and specifically YouTube. Nearly 1M hours each of Speech and Video data from this source far outstrips the next most common sources, which comprise less than 100K hours. For Speech, the primary data sources used to be Calling Platforms (pre-2017), content manually collected with Human Participation, and Audiobooks, but since 2018 internet videos have supplanted these other sources. For Video, since 2013, YouTube, synthetic, and general web data sources all constitute a significantly larger portion of data used in prominent video datasets, outstripping the use of Movies, Flickr, Getty, or human curated sources. Among text post-training datasets, we see a similar trend with general or news web-based sources, including encyclopedic sources (mainly Wikipedia), providing the majority of tokens over time. Encyclopedic sources alone now contribute over 1T tokens in total.

**Synthetic data sources are rising the most rapidly.** Within the video modality, the introduction of VidProM (Wang & Yang, 2024a) in 2024, consisting of nearly 7M synthetically generated videos, offered a large shift in the video source distribution. Within the textual modality, from fig. 1, synthetic data represented <0.1% of the quantity of Web Encyclopedia data in 2020, but is now 10% its proportion in 2024, making up the 5th largest source of tokens. The top models used in generating datasets are mainly from OpenAI. The top 5 consist of ChatGPT, version unspecified (15.0% of synthetic datasets), GPT-4 (14.4%), BART (10.1%), GPT-3 (8.3%) and GPT-3.5-Turbo (4.9%). The average synthetic dataset also has notably longer turns (in tokens) than the average natural dataset: 1,756 tokens vs 1,065. The task distribution of textual synthetic datasets is shifted towards longer form, open-generation and creative tasks. For example, 88.1% of natural datasets contain classification tasks, compared to only 66.3% of synthetic datasets. Natural data is also more likely to cover translation than synthetic data (72.4% of datasets vs only 22.9% of synthetic datasets).

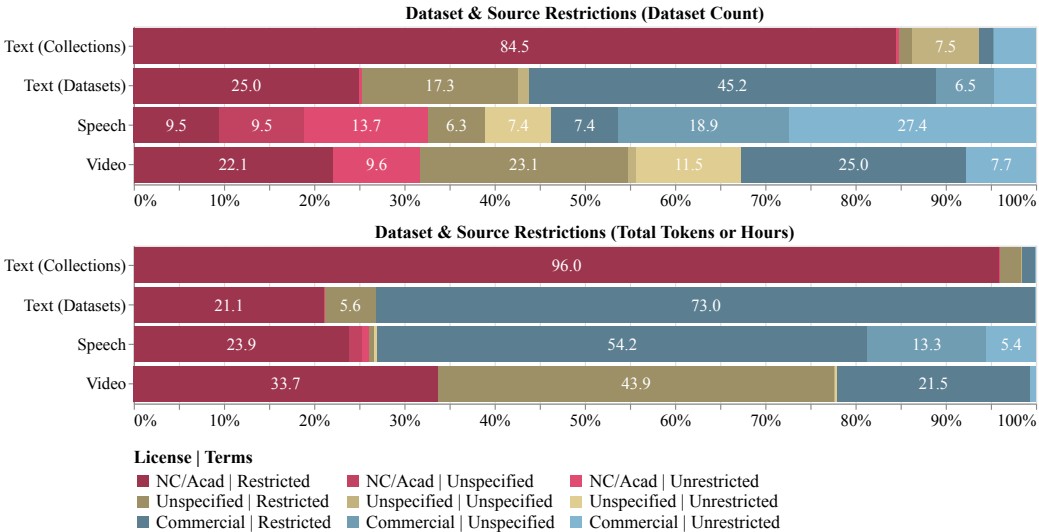

Figure 2: The distribution of restrictions from dataset *licenses* and their sources' *terms*. We break this down by the count of datasets (top), as well as total tokens or hours (bottom). Each license is categorized as Non-commercial/Academic (NC/Acad), Unspecified, or Commercially licensed. Each dataset may also have terms from the source: Restricted to non-commercial use, Unspecified restrictions, or Unrestricted. **Two main findings across modalities emerge: (1) Commercially licensed datasets represent a larger set of tokens and hours, relative to number of datasets; however, (2) the vast majority of those commercially licensed tokens/hours bare restrictions from their sources.** Tables 3 and 4 in the appendix provide detailed numbers.

## 3.2 Inconsistent Use Restrictions

In the United States, creators of a work automatically have a copyright interest that gives them exclusive rights to make copies and derivatives of the work (17 U.S.C. § 106). *Licenses* are legal documents through which the owners of a work express how others may use their work. By contrast, *Terms of Service* are a contract between a platform and its users governing how a platform and its content may be used (Robinson & Zhu, 2020). For simplicity, we use *"Licenses"* to refer to dataset restrictions, and *"Terms"* to refer to restrictions on the sources of datasets. There remain open questions about whether certain data licenses are enforceable, but these licenses signal the intention of data creators and therefore warrant consideration as the data creators may be best positioned to understand the sensitivities of the data (privacy, copyright, representation, etc.), and the most impacted by its downstream use (Morton-Park, 2023; Lee et al., 2023b; Mahari & Longpre, 2023; Mahari et al., 2023). How closely practitioners adhere to dataset licenses or source terms remains an open question, and may depend on jurisdiction or the desired model's use cases (Lee et al., 2023b). *This work does not propose one standard for all developers.* For these reasons we restrict our treatment and discussion here to tracing the lineage and distribution of licenses and terms for a given modality.

**Data source terms are much more restrictive than the dataset's documented license restrictions.** In Figure 2, we find only 25%, 33%, and 32% of text/speech/video datasets are licensed non-commercially. This value is even lower if we consider the proportion of tokens or hours, with 21%, 26%, and 33% of text/speech/video quantities carrying license restrictions. However, a staggering 99.8%, 78%, and 99% of those quantities carry some form of non-commercial restriction on one of their sources. For text, these restrictions are frequently from being generated by OpenAI or other models with a non-compete clause, while for speech and videos this is often since the datasets are derived from web or social media sources.

**Inconsistencies between dataset licenses and their source's restrictions pose challenges to practitioners.** A large amount of datasets have permissive or unspecified licenses, but some set of their sources carry non-commercial restrictions. This inconsistency is measurable—representing 79% of tokens in text datasets, 55% of speech hours, and 65% of video hours. Additionally, 19%, 14%, and 36% of text, speech, and video datasets have no license or intended use documentation

(from our audit of the datasets' documentation on Hugging Face Datasets, GitHub, and Papers with Code). A lack of centralized documentation around these restrictions means it can be misleading to developers who are attempting to source data according to their own legal standards for copyright and privacy. Furthermore, lack of documentation can hamper developers following best practices around data preparation and transparency (Gebru et al., 2021; Bommasani et al., 2023).

**Large quantities of commercially licensed text datasets are locked in collections without clear information to separate them from restrictive datasets.** In Figure 2 (top and bottom), we see the number of datasets and number of tokens *without* restrictions is significantly higher for Text (Datasets) than Text (Collections). Specifically, 60% more Datasets (or 75% more tokens) are commercially licensed, than for Collections. This demonstrates that many collections contain significant amounts of commercially licensed data. While our audit traces licenses for all datasets within a collection, most collections do not aggregate or expose this documentation. As a result, practitioners may be left without easy access to filter for the subsets appropriate for their sourcing standards.

### 3.3 GEOGRAPHICAL & LINGUISTIC REPRESENTATION IS NOT IMPROVING

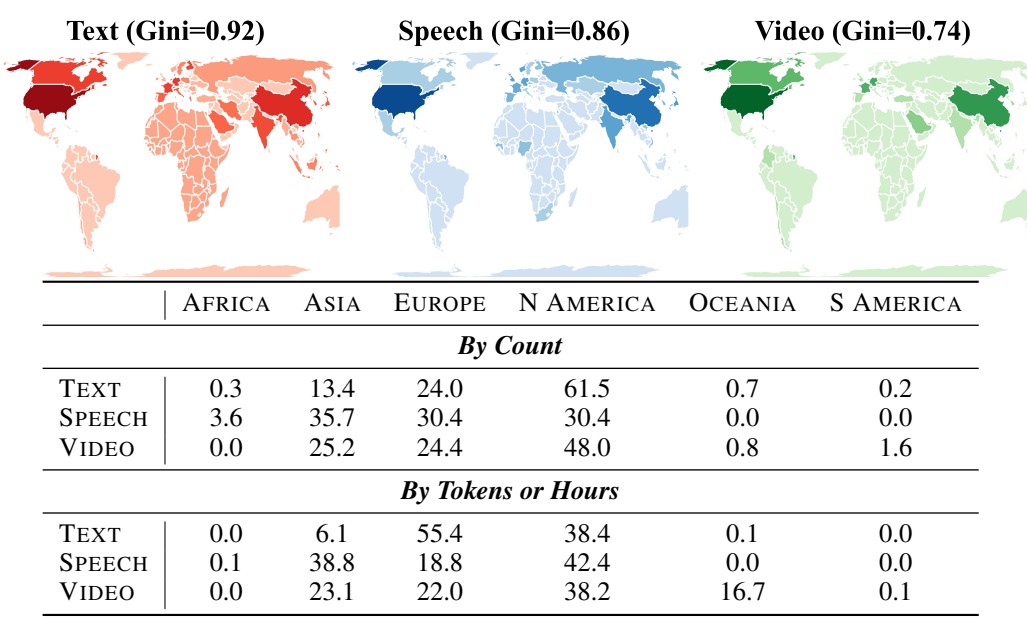

|  | AFRICA | ASIA | EUROPE | N AMERICA | OCEANIA | S AMERICA |
|---|---|---|---|---|---|---|
| ***By Count*** | | | | | | |
| TEXT | 0.3 | 13.4 | 24.0 | 61.5 | 0.7 | 0.2 |
| SPEECH | 3.6 | 35.7 | 30.4 | 30.4 | 0.0 | 0.0 |
| VIDEO | 0.0 | 25.2 | 24.4 | 48.0 | 0.8 | 1.6 |
| ***By Tokens or Hours*** | | | | | | |
| TEXT | 0.0 | 6.1 | 55.4 | 38.4 | 0.1 | 0.0 |
| SPEECH | 0.1 | 38.8 | 18.8 | 42.4 | 0.0 | 0.0 |
| VIDEO | 0.0 | 23.1 | 22.0 | 38.2 | 16.7 | 0.1 |

Figure 3: The geographical distribution of countries (world maps) and continents (table) represented by dataset creators. **Despite some differences in European, Russian, and Middle Eastern representation, creators are heavily concentrated in the US, China, and Western Europe, with little to no representation in South America or Africa, across modalities.** The current Gini coefficient for (Text, Speech, Video) = (0.92, 0.86, 0.74), where higher values indicate more concentration.

**The importance and progress of representation in AI training data.** Diversity and representation in training datasets, and among their creators, are widely acknowledged as essential to building AI models that are less biased, more useful, and more equitable (Joshi et al., 2020; Singh et al., 2024a; Üstün et al., 2024; Adelani et al., 2021; 2024; Aakanksha et al., 2024; McMillan-Major et al., 2022b; Porgali et al., 2023; Monfort et al., 2019a; Sigurdsson et al., 2016a). Prior work has measured cultural, ideological, geographic and linguistic imbalances in data (Faisal et al., 2022; Shankar et al., 2017; McMillan-Major et al., 2022a; De Vries et al., 2019; Mahadev & Chakravarti, 2021). These studies have exposed significant flaws, often in the form of bias and discrimination, stemming directly from poor representation in data (Buolamwini & Gebru, 2018b; Birhane et al., 2021). As this problem has now been widely acknowledged for decades, recent efforts have foregrounded sourcing data multilingually and multi-culturally, from native speakers and creators (e.g. ROOTS (Laurençon et al., 2022), the Aya Dataset (Singh et al., 2024a), the SEACrowd Catalogue (Lovenia et al., 2024), the Masader Catalogue (Alyafeai et al., 2022), Common Voice (Ardila et al., 2019), Causal Conversations V2 (Porgali et al., 2023) or Moments in Time (Monfort et al., 2019a)).

**Measuring geographical and linguistic representation.** Naturally, we aim to use our audit to measure the progress of these efforts on geographical and linguistic representation in the AI ecosystem. We measure the progress of two forms of representation: (1) language diversity of text and speech data, and (2) geographical diversity of the creators, in all three modalities. For languages, we use the ISO 639-1 and 639-3 language codes and top-level language families from Glottolog 5.0. In Figure 4(a, c) we display the cumulative sum of unique languages and countries present across all audited datasets, at each time period since 2013. While these measurements illustrate the absolute rise in diversity, we also hope to measure the relative dispersion, or equality of languages and countries in the distribution. In Figure 4(b, d), we use the Gini Index (Wilson, 1914; Atkinson et al., 1970), a traditional measure of statistical dispersion, frequently used to quantify inequality. This allows us to understand if the distributions of languages and creators are more representative of the international community over the last decade, or equally concentrated despite apparent efforts at the margins.

**Inequality in geographical representation remains very high, with few organizations creating datasets from the Global South.** For every dataset, our audit recorded the organizational affiliations of each creator of the dataset.[5] These organizations were then manually mapped to the country in which they are headquartered. Occasionally, organizations like BigScience, BigCode, or Masakhane have international or continental representation, and were counted as such. In Figure 3, we measure the current state of diversity among these creator organizations—where a Gini coefficient of 1 indicates highest concentration, and lower values more broad representation. Without taking up the normative question of what a truly "fair" score would be, these values provide useful comparisons across modalities and over time. We find that Text dataset developers are particularly homogeneous, with a Gini-coefficient of 0.92; followed by Speech, at 0.86 and Video at 0.74, which remain high, but are meaningfully less concentrated. Figure 3 also illustrates that even this limited diversity is still concentrated in North America, Europe, East Asia, and less so in the Global South.

In Figure 3, we also compare the distribution of datasets, and of tokens or hours by continent. Dataset creators affiliated with African or South American organizations account for fewer than 0.2% of all tokens or hours, in each modality. In contrast, Asian affiliated organizations represent large proportions of the data, particularly for speech (39% of hours, attributed predominantly to YODAS (Li et al., 2023b)). Much of this driven by Chinese, Indian, Russian, and Saudi Arabian creators. Most prominently, the combination of North American and European datasets comprises 93% of text tokens, 61% of speech hours, and 60% of video hours.

**Geographical representation has not significantly improved for over a decade.** In Figure 4(c), we measure the total unique number of countries represented across all dataset creator organizations. While individual creators will have varying ethnic and national affiliation, we treat this as an estimate for the influence of each locale in dataset development. We find that while the number of represented countries has risen steadily each year, for each modality, this represents only an illusion of progress. Empirically, the Gini coefficient for each modality has not significantly changed since the start of the period we examine in 2013. Geographic diversity has increased only among Video datasets, and these increases are not significant at the $p = 0.05$ level. Text and Speech geographical representations appear to remain stable over the last decade of AI development.

**Multilingual representation has not improved by most measures.** Similar to geographical representation, we measure the cumulative number of ISO 639-1 languages and language families over time, as well as the per-modality Gini-coefficient. Figure 4(a) shows significant increases in the number of languages available for speech and text, especially in 2019, and 2023, with the introduction of large sets like Flores (Goyal et al., 2022), xP3x (Muennighoff et al., 2023), Common Voice (Ardila et al., 2019), and the Aya Collection (Singh et al., 2024a). However, once again, when measuring the cumulative dispersion of these datasets in Figure 4(b), only Text language families demonstrate any improvement from pre-2013 to the present. Improvements in the Gini coefficient appear to be largely driven by individual large-scale projects like xP3x and Common Voice, both introduced in 2019. Subsequently, newer datasets remain predominantly monolingual, causing measures of concentration in text languages, speech languages, and language families to remain consistently high.

**Academia, research non-profits, and industry labs continue to drive public dataset development.** We manually categorize the organizations creating popular datasets into: Academic Organization

---

[5]A dataset creator, following (Longpre et al., 2024b), is defined as an organization associated with the release of the dataset as created for machine learning—not any of the upstream sources. More details in Appendix D.

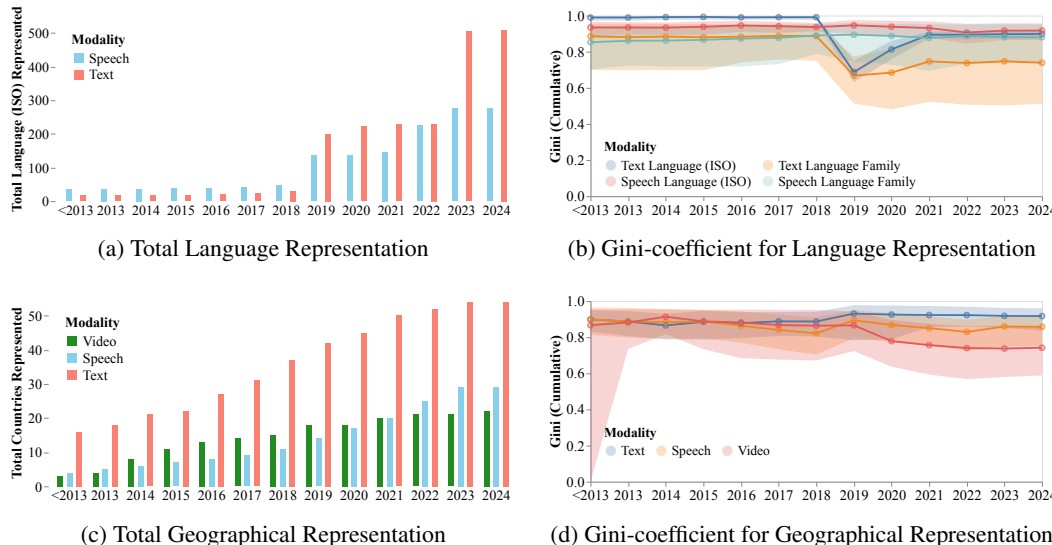

(a) Total Language Representation

(b) Gini-coefficient for Language Representation

(c) Total Geographical Representation

(d) Gini-coefficient for Geographical Representation

Figure 4: The cumulative totals (left) of languages and countries represented in the data over time, and the 95% confidence intervals of the gini-coefficients over time (right) to measure the representativeness of these variables. Gini-coefficients are a measure of statistical dispersion, frequently used to quantify inequality. A Gini coefficient of 1 indicates highest concentration, and lower values more broad representation. **While the number of represented languages and geographies continue to rise (left), the equality of their distribution has in most cases, not significantly changed.**

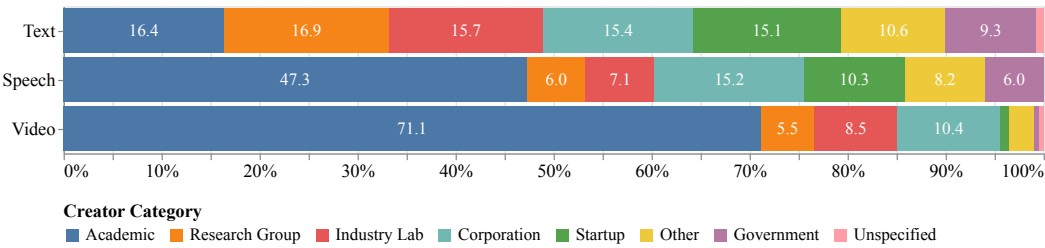

Figure 5: The distribution of creator organizations by modality. **Most public speech and video datasets are developed by academic organizations, whereas text datasets are developed by a wide mix of academia, non-profit or industry labs, as well as startups.**

(e.g., universities), Research Groups (e.g., non-profits like BigScience, EleutherAI and AI2), Industry Labs (e.g., Cohere for AI, Google DeepMind), Corporations (e.g. Google, Meta), Startups (e.g., OpenAI, Anthropic), Governments, Unspecified (owner affiliation not shared), and Other. When a dataset is released in collaboration between organizations, we record each organization. In Figure 5, we find that universities and other academic organizations account for 16%, 47%, and 71% of all recorded dataset releases, across Text, Speech, and Video respectively. Research groups, industry labs and even corporations are also significant contributors, especially for Text datasets, where ecosystem contributors are far more distributed. Academic organizations' greater role in Video and Speech may suggest special impediments (e.g., privacy-related) to commercial releases of these datasets.

## 4 DISCUSSION

**The rise of web, social media, and synthetic data may pose greater privacy, copyright, and bias risks.** Section 3.1 discusses the rise of web content and particularly social media as primary sources for speech and video. Figure 1 shows these sources now exceed by at least an order of magnitude more traditional, curated sources like movies, audiobooks, radio, TV, or data produced by human participants. Their mostly user-generated content makes these websites a natural source for the quantity, freshness, and heterogeneity needed to train general-purpose models (Longpre et al.,

2023; Aghajanyan et al., 2023). However, prior work suggests that user-generated web data is also more challenging to use than curated data, particularly for privacy, copyright, bias, and factuality. Web-based and particularly user-generated content is disproportionately likely to include personally identifiable information (PII) Luccioni & Viviano (2021); Subramani et al. (2023); Elazar et al. (2023), and copyrighted content (Meese & Hagedorn, 2019; Lee et al., 2023b). These can be reproduced in the outputs of AI models (Carlini et al., 2022; Chen et al., 2023c), creating privacy and copyright concerns (Zhang et al., 2023a). Open datasets being used to train GPAI often attempt to filter—but frequently miss—PII and copyrighted data (Soldaini et al., 2024; Subramani et al., 2023) (although not all do (Penedo et al., 2023)). Social media, in particular, is also known to have bias, toxicity and factuality issues (Olteanu et al., 2019), which can manifest in trained models, even after alignment (Kotha et al., 2023). Lastly, while synthetic data can help reduce the prevalence of PII, copyright, or bias in data, it comes with its own challenges (Kurakin et al., 2023; Liu et al., 2024a).

**Social media sites are a very prominent data source, but their terms often restrict crawling or commercial use.** We find 71% of Video and 69% of Speech data to be from YouTube, whose scale, freshness, and multimodality (containing videos, speech, images, and text) have made it a prominent data source (Abu-El-Haija et al., 2016b; Aytar et al., 2018; Chang et al., 2020; Uthus et al., 2023; Coats, 2023; Li et al., 2023b). However, YouTube is a social media platform owned by Google and its Terms of Service[6] prohibit third-party crawling. While content creators retain ownership of material uploaded to YouTube, the YouTube terms of service also grant Google a license to reproduce, modify, display, and use the content for YouTube's "business" (which may include building machine learning models), and forbid crawling by third parties, even if the copyright holder has selected a permissive license. Model developers like Nvidia and OpenAI have been sued in the U.S. by content creators who allege that they unlawfully trained on YouTube videos (Cole, 2024; Skolnik, 2024). Large social media sites like Reddit and StackOverflow[7] have also recently adopted restrictive terms, with access being restricted just as these data sources become critical to scaling AI systems. The enforceability of these licenses and terms, however, is an open legal question beyond the scope of our work.

**Ambiguous and poorly documented use restrictions may significantly inhibit model developers adhering to cautious legal and ethical data sourcing standards.** In Section 3.2. we find that a significant amount of data carry non-commercial restrictions in their sources, rather than on the final dataset, which can contain no license or a permissive one. For text and video, these restrictions can reach 99% of all tokens and hours. These inconsistencies are the result of datasets being iteratively re-packaged and re-licensed, without carrying on documentation (Longpre et al., 2024b). While not every developer will employ the same filtering standards, our work shows that separating and identifying appropriate datasets remains difficult across these modalities. Without continued audits and documentation, practitioners may have to either take on avoidable risk or forgo large collections of partially viable data, hampering data scaling laws (Kaplan et al., 2020). We hope our audit will help practitioners apply their own standards and make informed decisions on training data use.

**The limitations of measures of geographical and linguistic representation.** Measures of geographic and linguistic representation are imperfect: We have only partial information about the developers' identities (including for privacy reasons), limited transparency into how frequently these datasets are used, and cannot say how far proprietary datasets may fill in representation gaps behind closed doors. Nonetheless, we believe the breadth and rigour of the audit make this the best available empirical measure of representation in *public* datasets. Indeed, we contend that it is necessary to measure representation in AI data to understand progress, or its absence, towards AI systems that fairly serve the broad user community. Figures 3 and 4 show that despite the absolute rise of geographic and linguistic representation, a western-centric skew persists across thousands of surveyed datasets. We release all audit materials for transparency, replicability, and further research use.

**Conducting representative analyses of an ecosystem comes with assumptions.** The AI ecosystem is decentralized. Text datasets, for example, are often hosted on HuggingFace, unlike Speech and Video. Similarly, while Text data is frequently repackaged for general-purpose post-training, this is less true of other modalities. Scoping and dataset selection thus need to be modality-specific, rather than a single general protocol unable to capture these nuances. Similarly, we studied modalities of interest to foundation model development (Brooks et al., 2024; Radford et al., 2023), but many others remain for future work (e.g., images, 3D shapes, tabular, time series, graphs, and geospatial data).

---

[6]YouTube Terms of Service.
[7]Reddit User Agreement and StackOverflow Terms of Service.

ACKNOWLEDGMENTS

This research was conducted by the Data Provenance Initiative, a collective of independent and academic researchers volunteering their time to data transparency projects. The Data Provenance Initiative is supported by the Mozilla Data Futures Lab Infrastructure Fund.

## A  EXTENDED RELATED WORK

Progress in machine learning across modalities from speech (Radford et al., 2023) to vision (Dosovitskiy et al., 2021) to text (Brown et al., 2020a; Wei et al., 2021) has benefited from advancements in large pre-training and fine-tuning corpora. The development of multimodal corpora has also been key to several recent advances, as with CLIP in the image/text domain Radford et al. (2021), CLAP for audio/text settings Elizalde et al. (2022), and a number of other models involving both text and images, audio or video (Radford et al., 2023; Ramirez et al., 2024; Singer et al., 2022; Ramesh et al., 2022).

The datasets powering these advances are not, however, always well-documented, despite the existence of standards and frameworks for recording and annotating dataset metadata that range from 'data statements' (Bender & Friedman, 2018) to 'datasheets for datasets' (Gebru et al., 2021) and others (Mitchell et al., 2019). The key problem is not a deficiency of any particular framework, but rather inconsistent adoption and fragmentation (Longpre et al., 2024d). Much prior work has argued for the need to document and audit these datasets (Rogers, 2021; Paullada et al., 2021), motivated by concerns from reproducibility (Kapoor & Narayanan, 2022) to interpretability (Longpre et al., 2023) to bias and fairness problems that may stem from problematic content in training data (Birhane et al., 2021).

There have been several attempts to carry out such audits, with prior work examining pretraining data (Longpre et al., 2024c), general web corpora (Gao et al., 2020; Dodge et al., 2021), instruction fine-tuning datasets (Longpre et al., 2024b), and the documentation fields of the HuggingFace Datasets platform in particular (Yang et al., 2024). For speech and vision, there has been less work, with many discussions of datasets in the aggregate occurring in survey papers (Schiappa et al., 2023; Chaquet et al., 2013), research aimed directly at improving model performance Gadre et al. (2023) or close examinations of questions like bias in small groups of datasets (Buolamwini & Gebru, 2018b; Romanou et al., 2024).

Prior work has also examined the identities, affiliations and national origin of paper authors (Movva et al., 2024) in AI, but an analogous look at the producers of datasets is lacking. We aim to carry out such analyses: replicating those for pretraining and text finetuning datasets in video and audio domains, and surveying provenance and legal status. Finally, there has also been significant recent attention to legal questions in the collection and use of AI training data (Sag, 2020; Henderson et al., 2023). The complex process involved in preparing these datasets (Lee et al., 2023b), and the ambiguous licensing of inputs, can make understanding the legal status of the final output quite difficult.

## B  DATASET LICENSES & TERMS

**Detailed taxonomy**  We code the legal restrictions placed on use of datasets along two axes. First, we identify whether a dataset's license permits commercial use ("Commercial" in Table 3), only non-commercial / academic use ("NC / Acad"), or does not clearly specify what is permitted ("Unspecified"). The latter category includes datasets for which we were unable to locate a license. Datasets which are in the public domain and not subject to a license are counted as commercially usable. Second, we annotate the contractual or terms-of-use restrictions placed on dataset use by the source of each dataset. There are four levels, defined in Table 3. Note that the Model Closed status can only apply to datasets that are AI-generated, at least in part. Some datasets can carry both Model Closed and Source Closed status, but we count the Model Closed first for simplicity.

**Detailed breakdown**  Tables 3 and 4 present crosstabs of these two dimensions, according to respectively the total amount of content and the number of datasets. The most notable finding, as discussed in the main text, is the frequency of clashing restriction status between licenses and terms.

| LABEL | DEFINITION |
|---|---|
| MODEL CLOSED | A model used to generate part or all of the dataset prohibits using its outputs commercially, to develop a competing AI model, or in general. |
| SOURCE CLOSED | The source has a license or terms that prohibits use of the data, either commercially, from being crawled, to develop AI, or in general. |
| UNSPECIFIED | No information can be found relevant to restrictions, or lack thereof, for this source. |
| UNRESTRICTED | The source has a commercially permissive license, such as CC BY, or explicitly states the data is open for broad use. |

Table 2: **The taxonomy used to determine use restrictions on each dataset source.** Each source in a dataset is examined and fit into one of these categories. The dataset Terms are then labelled according to the strictest terms across the sources, with Model Closed and Source Closed considered stricter than Unspecified which is in turn stricter than Unrestricted.

By amount of content, fully 73.0% of text content, 55.0% of speech content, and 21.6% of video content is subject to a license permitting commercial use but also to terms restrictions forbidding it, or the reverse. The absolute level of restrictions is also high, with < 0.1% of text content, 5.4% of speech content, and 0.6% of video content usable for commercial purposes under both licenses and terms.

| LICENSE / TERMS | RESTRICTED | UNSPECIFIED | UNRESTRICTED | TOTAL |
|---|---|---|---|---|
| *Text Collections* | | | | |
| NC/ACAD | 96.0 | 0.0 | 0.0 | 96.0 |
| UNSPECIFIED | 2.3 | 0.1 | 0.0 | 2.4 |
| COMMERCIAL | 1.5 | 0.0 | 0.0 | 1.6 |
| TOTAL | 99.8 | 0.1 | 0.1 | |
| *Text Datasets* | | | | |
| NC/ACAD | 21.1 | 0.0 | 0.0 | 21.2 |
| UNSPECIFIED | 5.7 | 0.1 | 0.0 | 5.7 |
| COMMERCIAL | 73.0 | 0.0 | 0.0 | 73.1 |
| TOTAL | 99.8 | 0.1 | 0.1 | |
| *Speech Datasets* | | | | |
| NC/ACAD | 23.9 | 1.4 | 0.8 | 26.2 |
| UNSPECIFIED | 0.5 | 0.0 | 0.4 | 0.9 |
| COMMERCIAL | 54.2 | 13.3 | 5.4 | 73.0 |
| TOTAL | 78.6 | 14.7 | 6.7 | |
| *Video Datasets* | | | | |
| NC/ACAD | 33.7 | 0.0 | 0.1 | 33.8 |
| UNSPECIFIED | 43.9 | 0.1 | 0.1 | 44.1 |
| COMMERCIAL | 21.5 | 0.0 | 0.6 | 22.1 |
| TOTAL | 99.1 | 0.1 | 0.8 | |

Table 3: **A breakdown of the percentage of license and terms restrictions across datasets**, by total tokens or hours of content. The much higher frequency of restrictions at the collection level is because we consider a collection's license or terms status to be the most restrictive of those for its datasets. Note that percentages may not add to exactly 100% because of rounding.

| LICENSE / TERMS | RESTRICTED | UNSPECIFIED | UNRESTRICTED | TOTAL |
|---|---|---|---|---|
| *Text Collections* | | | | |
| NC/ACAD | 84.5 | 0.0 | 0.3 | 84.8 |
| UNSPECIFIED | 1.5 | 7.5 | 0.0 | 8.9 |
| COMMERCIAL | 1.5 | 0.2 | 4.5 | 6.3 |
| TOTAL | 87.5 | 7.7 | 4.8 | |
| *Text Datasets* | | | | |
| NC/ACAD | 25.0 | 0.0 | 0.3 | 25.3 |
| UNSPECIFIED | 17.3 | 1.2 | 0.0 | 18.5 |
| COMMERCIAL | 45.2 | 6.5 | 4.5 | 56.2 |
| TOTAL | 87.5 | 7.7 | 4.8 | |
| *Speech Datasets* | | | | |
| NC/ACAD | 9.5 | 9.5 | 13.7 | 32.6 |
| UNSPECIFIED | 6.3 | 0.0 | 7.4 | 13.7 |
| COMMERCIAL | 7.4 | 18.9 | 27.4 | 53.7 |
| TOTAL | 23.2 | 28.4 | 48.4 | |
| *Video Datasets* | | | | |
| NC/ACAD | 22.1 | 0.0 | 9.6 | 31.7 |
| UNSPECIFIED | 23.1 | 1.0 | 11.5 | 35.6 |
| COMMERCIAL | 25.0 | 0.0 | 7.7 | 32.7 |
| TOTAL | 70.2 | 1.0 | 28.8 | |

Table 4: **A breakdown of the percentage of license and terms restrictions** by dataset count. The much higher frequency of restrictions at the collection level is because we consider a collection's license or terms status to be the most restrictive of those for its datasets. Note that percentages may not add to exactly 100% because of rounding.

## C  ADDITIONAL RESULTS

Figures 6 and 7 report the size distributions of the datasets. We measure size differently for different types of datasets: Text datasets are in tokens, and audio/video in hours of content. The lack of standard tokenization or preprocessing schemes for those modalities makes it simplest to report raw dataset size.

Notably, we find quite different size distributions by modality. The distribution of dataset sizes has the thickest right tail for text, followed by speech and then by video. Most video datasets are short in hour terms, with speech datasets tending to be somewhat longer and text datasets having a greater prevalence of both very small and very large datasets relative to the mean size.

Dataset tasks, meanwhile, reflect traditional approaches and research programs for each modality. Classification is the most common task for both text and video, with the video community's long-standing interest in captioning also visible in its role as the second most common task for video datasets. Q&A occupies a similar role for text, though text datasets have a more balanced distribution over other, increasingly prominent tasks like generation and reasoning. Given our selection criteria, all datasets for speech are for ASR tasks, but other tasks like speaker identification and translation are also represented.

## D  DATASETS

This section provides a detailed overview of the datasets we have collected and analyzed. Table 5 summarizes the text datasets, Table 6 the audio datasets, and Table 7 the video datasets. Each of these

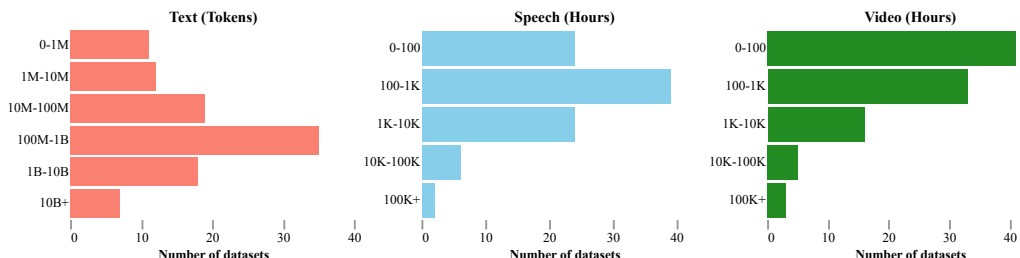

Figure 6: The distribution of dataset sizes for each modality. Most text data collections are between 100M-1B tokens. **Speech datasets average 100-1k hours, and video datasets are usually the smallest, commonly less than 100 hours.**

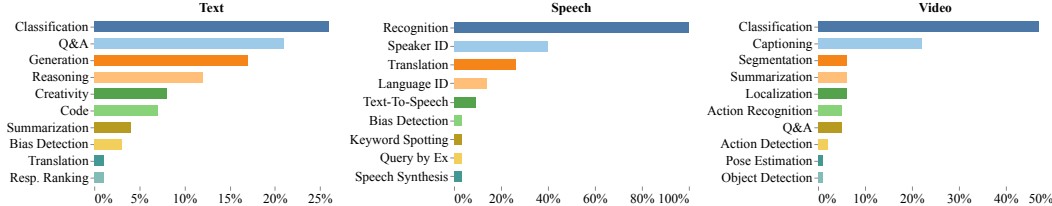

Figure 7: The task distribution of datasets, across modalities. Post-training text and video datasets are predominantly based on classification. For text, generation and reasoning are rising categories. All speech datasets are recognition-based, particularly for speaker, language, or in the process of translation.

tables lists broad collections of data, sorted in chronological order, and provides information about their properties, sizes, sources and permissions. Each collection can include multiple datasets, and they generally reflect the ways dataset creators have grouped their datasets (such as in the same paper). Because of the large number of datasets, we provide detailed information about their licenses and original published papers, where applicable, in the supplementary Attribution Card in Appendix F.

**Annotation Details: Text**   For post-training text datasets it is common to package many together as collections, such as Flan (Wei et al., 2021) or P3 (Sanh et al., 2021). This practice is not common to the same extent for speech or video datasets. For much of the text analysis, where possible, we chose to analyze statistics at the collection-level, since practitioners are more likely to adopt a collection for general-purpose post-training, than an individual dataset within the collection. Also, in dataset-level statistics, metadata for a single collection with many datasets can get repeated and overwhelm the statistics unfairly (e.g. the dataset aggregator/creator being repeated hundreds of times). Consequently, our collection-level analysis of the text modality is reflected in Figure 1, Figure 3, Figure 5, Figure 4, Figure 7, and Figure 6. However, for Figure 2 we draw the distinction between collection and dataset metrics, as practitioners may wish to unpack collections to extract only commercially licensed data. In that case a Collection inherits the most restrictive license and terms of its constituent datasets.

For annotating creator organizations, we follow prior work's instructions (Longpre et al., 2024b). For each dataset they record the affiliations listed on the academic paper or GitHub or HuggingFace object in which the dataset was released. This does not include the organizations who created or owned the sources from which the data was derived. For instance, the SQuAD dataset (Rajpurkar et al., 2016a) would be associated with Stanford (the authors' affiliation), but not Wikipedia, which the data was partially derived from. For a dataset that has authors affiliated with multiple organizations, the dataset will be counted towards each organization.

**Annotation Details: Speech**   In many cases, multiple versions of a dataset exist due to datasets being expanded or updated. In these scenarios, we used the release date from the initial version (since release dates for subsequent versions were not always clear), but used metadata from the most recently released version for which information was available to offer an overview of the current landscape of data. However, if the dataset versions could not be meaningfully aggregated (e.g. different licenses),

or did not appear to be cumulatively designed (non-overlapping or otherwise semantically disjoint data), we maintained separate records. We kept only datasets for which ASR was noted as a primary task. For example, if a dataset was primarily intended for text-to-speech or speaker recognition, we did not keep it even if it could conceivably be repurposed for ASR. When computing hours, we excluded any hours without supervisory transcripts/scripts (unlabeled data), but kept hours with "weak supervision" (e.g. model-generated transcripts from speech audio). We recognize the difficulty in comprehensively covering all relevant datasets.

**Annotation Details: Video**  In video, a single dataset can be re-purposed and annotated to address different tasks Monfort et al. (2019a; 2021a). We consider these as two different datasets even if they have the same video source since now they can be used for different computer vision tasks.

Table 5: **Alignment tuning (text) collections and properties**. Collection properties include numbers of datasets, tasks, languages, and text domains. The SOURCE column indicates whether a collection contains human-generated web text (🌐), language model outputs (🤖) or both (🌐🤖). The USE column indicates whether a collection includes data freely usable even for commercial purposes (●), data usable only for noncommercial purposes or academic research (●) and data whose license status is not specified precisely enough to allow us to determine commercial use permissions (●). Note that each collection may have different datasets with one, two, or all three of these statuses. Finally, the OAI column indicates collections which include OpenAI model generations. Datasets are sorted chronologically to highlight trends over time.

| COLLECTION | | PROPERTY COUNTS | | | | TYPES | PERMISSIONS | |
| --- | --- | --- | --- | --- | --- | --- | --- | --- |
| | YEAR | DATASETS | TASKS | LANGS | DOMAINS | SOURCE | USE | OAI |
| RiddleSense | 2021 | 1 | 3 | 1 | 1 | web | ● | |
| MathInstr. | 2023 | 1 | 3 | 1 | 1 | LM | ● | ✔ |
| No Robots | 2023 | 1 | 8 | 1 | 1 | web | ● ● | |
| Nectar | 2023 | 1 | 1 | 1 | 2 | LM | ● ● | ✔ |
| MetaMathQA | 2023 | 8 | 2 | 1 | 1 | LM | ● | ✔ |
| MegaWika | 2023 | 50 | 1 | 50 | 1 | LM | ● | |
| MedInstr. | 2023 | 1 | 1 | 1 | 1 | LM | ● | ✔ |
| MathDial | 2023 | 1 | 2 | 1 | 4 | LM | ● | ✔ |
| PII-Masking-200k | 2023 | 1 | 2 | 4 | 1 | web | ● | |
| Pure-Dove | 2023 | 1 | 4 | 1 | 1 | LM | ● | ✔ |
| LMSYS-Chat-1M | 2023 | 1 | 9 | 5 | 1 | LM | ● ● | ✔ |
| PygmalionAI-PIPPA | 2023 | 1 | 3 | 1 | 1 | LM | ● | |
| HelpSteer | 2023 | 1 | 5 | 1 | 1 | web | ● | |
| SeaBench | 2023 | 9 | 4 | 9 | 5 | LM | ● | |
| Open Asst. v2 | 2023 | 19 | 4 | 19 | 1 | web | ● | |
| Feedback Coll. | 2023 | 1 | 2 | 1 | 1 | LM | ● | ✔ |
| Glaive Code Asst. | 2023 | 1 | 2 | 2 | 1 | LM | ● | |
| EverythingLM | 2023 | 1 | 8 | 2 | 1 | LM | ● | ✔ |
| Bactrian-X | 2023 | 6 | 4 | 6 | 1 | LM | ● ● | ✔ |
| COBRA Frames | 2023 | 1 | 1 | 1 | 2 | LM | ● | ✔ |
| UltraFeedback Argilla | 2023 | 9 | 16 | 1 | 20 | web+LM | ● ● ● | ✔ |
| ExpertQA | 2023 | 1 | 3 | 1 | 1 | LM | ● | ✔ |
| ChatDoctor | 2023 | 3 | 1 | 1 | 2 | web+LM | ● ● | ✔ |
| Capybara | 2023 | 11 | 17 | 2 | 1 | LM | ● ● ● | ✔ |
| UltraChat-200k | 2023 | 1 | 7 | 1 | 2 | LM | ● | ✔ |
| CollectiveCognition | 2023 | 1 | 6 | 1 | 1 | LM | ● | ✔ |
| Thai Gen AI | 2023 | 9 | 11 | 1 | 1 | LM | ● ● | ✔ |
| Deita 10K | 2023 | 2 | 11 | 1 | 3 | LM | ● ● | ✔ |
| SelFee | 2023 | 1 | 5 | 1 | 1 | LM | ● | ✔ |
| ChatbotArena | 2023 | 1 | 4 | 1 | 1 | LM | ● ● | ✔ |
| OpenGPT Healthcare | 2023 | 3 | 4 | 1 | 1 | LM | ● ● | ✔ |
| Orca-Math | 2024 | 1 | 1 | 1 | 3 | LM | ● ● | ✔ |
| OpenMathInstr.-1 | 2024 | 2 | 3 | 1 | 3 | LM | ● ● | |
| WildChat | 2024 | 2 | 7 | 10 | 1 | LM | ● | ✔ |
| Magpie-Pro | 2024 | 1 | 9 | 1 | 1 | LM | ● | |

Table 5: **Alignment tuning (text) collections and properties**.

| Collection | Year | Property Counts | | | | Types | Permissions | |
| | | Datasets | Tasks | Langs | Domains | Source | Use | OAI |
| --- | --- | --- | --- | --- | --- | --- | --- | --- |
| 10k Prompt Ranked | 2024 | 1 | 13 | 1 | 4 | [robot] | [tan] | ✔ |
| Synth.-GSM8K-Refl. | 2024 | 1 | 3 | 1 | 1 | [robot] | [blue] | |
| LongAlign-10k | 2024 | 1 | 3 | 1 | 1 | [robot] | [blue][red] | ✔ |
| Llama2-MedTuned-Instr. | 2024 | 1 | 4 | 1 | 1 | [globe] | [red] | |
| KIWI | 2024 | 1 | 1 | 1 | 2 | [robot] | [blue] | ✔ |
| Indic-Instr. | 2024 | 8 | 7 | 2 | 3 | [robot] | [blue][tan][red] | ✔ |
| Gretel Text-to-SQL | 2024 | 1 | 1 | 3 | 1 | [robot] | [blue] | |
| Conifer | 2024 | 1 | 8 | 1 | 2 | [robot] | [blue] | ✔ |
| Cidar | 2024 | 1 | 8 | 1 | 1 | [robot] | [red] | ✔ |
| Aya | 2024 | 71 | 7 | 71 | 1 | [globe] | [blue] | |
| Reasoning | 2024 | 1 | 4 | 1 | 1 | [robot] | [blue] | ✔ |
| AgentInstruct | Mult. | 6 | 3 | 1 | 7 | [globe][robot] | [blue][tan] | ✔ |
| InstAr | Mult. | 24 | 13 | 1 | 9 | [globe][robot] | [blue][tan][red] | ✔ |
| Dynosaur | Mult. | 1k | 21 | 1 | 22 | [globe][robot] | [blue][tan][red] | ✔ |
| Medical Meadow | Mult. | 8 | 2 | 1 | 3 | [globe][robot] | [blue][tan][red] | ✔ |
| Open-Platypus | Mult. | 10 | 10 | 36 | 8 | [globe][robot] | [blue][tan][red] | ✔ |
| PMC-LLaMA Instr. | Mult. | 7 | 1 | 1 | 2 | [globe][robot] | [blue][tan] | ✔ |
| COIG | Mult. | 18 | 13 | 2 | 22 | [globe][robot] | [blue][tan][red] | |
| DialogStudio | Mult. | 83 | 3 | 5 | 3 | [globe][robot] | [blue][tan][red] | |

Table 6: **Audio collections and properties**. Collection properties include numbers of audio hours (HR), speakers (SPKR), languages (LANG), creator institutions (CREAT), tasks (TASKS), data sources (SRC), and topics (TOPICS). The number of datasets is not listed because all collections include only one dataset, except for M2ASR which has four. The US column indicates datasets from or partly from the United States, the AC column datasets created by academic institutions, and the IND column datasets created by industry. Note that a dataset can have all of these, none of them, or any combination of them. The USE column indicates whether a collection includes data freely usable even for commercial purposes (🔵), data usable only for noncommercial purposes or academic research (🔴) and data whose license status is not specified precisely enough to allow us to determine commercial use permissions (🟤). Note that each collection may have different datasets with one, two, or all three of these statuses. Datasets are sorted chronologically to highlight trends over time.

| Collection | Year | Property Counts | | | | | | | Category | | | Perm |
| | | Hr | Spkr | Lang | Creat | Tasks | Src | Top | US | Ac | Ind | Use |
| --- | --- | --- | --- | --- | --- | --- | --- | --- | --- | --- | --- | --- |
| TIMIT | 1990 | 5 | 630 | 1 | 3 | 3 | 1 | 7 | ✔ | ✔ | ✔ | [red] |
| Switchboard | 1992 | 250 | 543 | 1 | 1 | 1 | 1 | 70 | ✔ | | ✔ | [red] |
| African Acc. French | 2003 | 22 | 232 | 1 | 1 | 1 | 1 | 7 | ✔ | | | [blue] |
| CSJ | 2003 | 661 | 1k | 1 | 1 | 1 | 1 | 2 | | | | [tan] |
| Fisher | 2004 | 2k | 12k | 1 | 1 | 1 | 1 | 36 | ✔ | ✔ | | [red] |
| CSLU 22 Langs. | 2005 | 84 | - | 21 | 1 | 1 | 1 | 7 | ✔ | ✔ | | [red] |
| AMI | 2005 | 100 | - | 1 | 1 | 1 | 2 | 2 | | ✔ | | [blue] |
| CSLU 1.2 | 2007 | 25 | 5k | 1 | 1 | 1 | 1 | 1 | ✔ | ✔ | | [red] |
| ALLSSTAR | 2010 | 86 | 140 | 27 | 1 | 1 | 1 | 3 | ✔ | ✔ | | [blue] |

Table 6: **Audio collections and properties**.

| Collection | Year | Hr | Spkr | Lang | Creat | Tasks | Src | Top | US | Ac | Ind | Use |
|---|---|---|---|---|---|---|---|---|---|---|---|---|
| TED-LIUM3 | 2012 | 452 | 2k | 1 | 2 | 2 | 1 | 1 | | ✔ | ✔ | ● red |
| NST Norwegian | 2013 | 540 | 870 | 1 | 1 | 1 | 1 | 7 | | | | ● blue |
| NST Danish | 2013 | 500 | - | 1 | 1 | 1 | 1 | 7 | | | | ● blue |
| NST Swedish | 2013 | 300 | - | 1 | 1 | 1 | 1 | 7 | | | | ● blue |
| Vystadial | 2014 | 56 | - | 2 | 1 | 1 | 2 | 3 | | ✔ | | ● blue |
| THCHS-30 | 2015 | 35 | 40 | 1 | 1 | 1 | 1 | 1 | | ✔ | | ● blue |
| LibriSpeech | 2015 | 1k | 2k | 1 | 1 | 1 | 1 | 106 | ✔ | ✔ | | ● blue |
| THUYG-20 | 2015 | 20 | 371 | 1 | 2 | 2 | 1 | 3 | | ✔ | | ● blue |
| VCTK | 2016 | 44 | 110 | 1 | 1 | 1 | 1 | 1 | | ✔ | | ● blue |
| Spoken Wikipedia | 2016 | 1k | 960 | 3 | 1 | 1 | 1 | 1 | | ✔ | | ● blue |
| AISHELL-1 | 2017 | 520 | 400 | 1 | 2 | 2 | 2 | 11 | | | ✔ | ● blue |
| LJSpeech | 2017 | 24 | 1 | 1 | 1 | 1 | 1 | 1 | ✔ | | | ● blue |
| ClarinPL | 2017 | 56 | 317 | 1 | 1 | 1 | 2 | 7 | | ✔ | | ● blue |
| AISHELL-2 | 2018 | 1k | 2k | 1 | 2 | 2 | 1 | 8 | | | ✔ | ● tan |
| Regional Af. Am. Lang. | 2018 | 159 | 222 | 1 | 1 | 1 | 1 | 8 | ✔ | ✔ | | ● red |
| Crowd Sourced Speech | 2018 | 1k | 3k | 5 | 1 | 1 | 1 | 1 | ✔ | | ✔ | ● blue |
| Zeroth-Korean | 2018 | 96 | 181 | 1 | 1 | 1 | 1 | 7 | | | ✔ | ● blue |
| RTVE | 2018 | 691 | - | 1 | 1 | 1 | 1 | 7 | | ✔ | | ● tan |
| OpenSTT | 2019 | 20k | - | 1 | 2 | 2 | 2 | 6 | | ✔ | ✔ | ● red |
| MuST-C | 2019 | 4k | 2k | 16 | 2 | 2 | 1 | 4 | | ✔ | | ● red |
| M-AILABS | 2019 | 1k | - | 8 | 1 | 1 | 1 | 33 | | | | ● tan |
| MAGICDATA | 2019 | 755 | 1k | 1 | 1 | 1 | 1 | 1 | | | ✔ | ● red |
| Common Voice 17 | 2019 | 31k | 330k | 124 | 3 | 3 | 1 | 1 | ✔ | ✔ | ✔ | ● blue |
| CoNASE | 2019 | 154k | - | 1 | 1 | 1 | 1 | 6 | | ✔ | | ● tan |
| Nigerian English | 2019 | 6 | - | 1 | 1 | 1 | 1 | 7 | ✔ | | ✔ | ● blue |
| Norwegian Parl. Speech | 2019 | 140 | 309 | 1 | 1 | 1 | 1 | 7 | | | | ● blue |
| 120h Spanish Speech | 2019 | 120 | 17 | 1 | 1 | 1 | 1 | 7 | | | | ● blue |
| DiDiSpeech | 2020 | 800 | 6k | 1 | 1 | 1 | 1 | 2 | | | ✔ | ● tan |
| Czech Parliament | 2020 | 444 | 212 | 1 | 1 | 1 | 1 | 7 | | ✔ | | ● blue |
| CoVoST-2 | 2020 | 3k | 78k | 22 | 1 | 1 | 2 | 1 | ✔ | | ✔ | ● blue |
| KSC | 2020 | 332 | - | 1 | 1 | 1 | 1 | 5 | | ✔ | | ● blue |
| Basq., Cat. and Gal. | 2020 | 34 | 132 | 3 | 1 | 1 | 1 | 2 | ✔ | | ✔ | ● blue |
| KsponSpeech | 2020 | 969 | 2k | 1 | 1 | 1 | 1 | 6 | | | | ● tan |
| Samromur | 2020 | 145 | 8k | 1 | 1 | 1 | 1 | 5 | | ✔ | | ● blue |
| Multiling. LibriSpeech | 2020 | 50k | 6k | 8 | 1 | 1 | 1 | 33 | ✔ | | ✔ | ● blue |
| MaSS | 2020 | 160 | - | 8 | 1 | 1 | 1 | 1 | | ✔ | | ● tan |
| FT SPEECH | 2020 | 2k | 434 | 1 | 2 | 2 | 1 | 2 | ✔ | ✔ | ✔ | ● tan |
| Eng. Acc. in Brit. Isles | 2020 | 31 | 120 | 1 | 1 | 1 | 1 | 4 | | | ✔ | ● blue |
| Highland Puebla Nahuatl | 2021 | 156 | - | 1 | 3 | 3 | 1 | 7 | ✔ | ✔ | | ● red |
| QASR | 2021 | 2k | 11k | 1 | 2 | 2 | 1 | 7 | ✔ | ✔ | ✔ | ● tan |
| Multiling. TEDx | 2021 | 765 | - | 9 | 3 | 3 | 1 | 7 | ✔ | ✔ | | ● red |
| Minds14 | 2021 | 25 | - | 14 | 1 | 1 | 2 | 7 | | | ✔ | ● blue |
| Golos | 2021 | 1k | - | 1 | 3 | 3 | 1 | 6 | | ✔ | ✔ | ● tan |

Table 6: **Audio collections and properties**.

| COLLECTION | YEAR | PROPERTY COUNTS | | | | | | | CATEGORY | | | PERM |
| | | HR | SPKR | LANG | CREAT | TASKS | SRC | TOP | US | AC | IND | USE |
|---|---|---|---|---|---|---|---|---|---|---|---|---|
| MASC | 2021 | 1k | 14k | 1 | 3 | 3 | 1 | 15 | | ✔ | ✔ | blue |
| LaboroTVSpeech | 2021 | 2k | - | 2 | 2 | 2 | 1 | 7 | | ✔ | ✔ | tan |
| KeSpeech | 2021 | 2k | 27k | 2 | 1 | 1 | 1 | 1 | | ✔ | | tan |
| JTUBESPEECH | 2021 | 1k | - | 2 | 4 | 4 | 1 | 7 | ✔ | ✔ | | tan |
| GigaSpeech | 2021 | 10k | - | 1 | 9 | 9 | 3 | 24 | ✔ | ✔ | ✔ | blue |
| VoxPopuli | 2021 | 2k | 4k | 16 | 1 | 1 | 1 | 1 | ✔ | | ✔ | blue |
| SPGISpeech | 2021 | 5k | 50k | 1 | 4 | 4 | 1 | 2 | ✔ | ✔ | ✔ | tan |
| West Afr. Radio | 2021 | 142 | - | 10 | 2 | 2 | 1 | 3 | ✔ | ✔ | | blue |
| AISHELL-4 | 2021 | 120 | 61 | 1 | 4 | 4 | 2 | 6 | ✔ | ✔ | ✔ | blue |
| West Afr. Virt. Asst. | 2021 | 2 | 49 | 3 | 2 | 2 | 1 | 2 | ✔ | ✔ | | blue |
| MediaSpeech | 2021 | 40 | - | 4 | 5 | 5 | 12 | 1 | | ✔ | ✔ | blue |
| People's Speech | 2021 | 30k | - | 1 | 7 | 7 | 2 | 14 | ✔ | ✔ | ✔ | blue |
| 1111 Hours Hindi | 2022 | 108 | - | 1 | 1 | 1 | 1 | 5 | | | ✔ | tan |
| Shrutilipi | 2022 | 6k | - | 12 | 2 | 2 | 1 | 1 | | ✔ | ✔ | blue |
| WenetSpeech | 2022 | 10k | - | 1 | 4 | 4 | 2 | 10 | | ✔ | ✔ | blue |
| Samromur Children | 2022 | 131 | 3k | 1 | 1 | 1 | 1 | 5 | | ✔ | | blue |
| SDS-200 | 2022 | 200 | 4k | 1 | 3 | 3 | 1 | 2 | | ✔ | ✔ | tan |
| aidatatang | 2022 | 200 | 600 | 1 | 1 | 1 | 1 | 7 | | | ✔ | red |
| Fleurs | 2022 | 1k | - | 102 | 3 | 3 | 1 | 11 | ✔ | ✔ | ✔ | blue |
| OLKAVS | 2022 | 1k | 1k | 1 | 2 | 2 | 1 | 14 | | ✔ | ✔ | tan |
| Norwegian Parl. | 2022 | 140 | 267 | 1 | 2 | 2 | 1 | 2 | | | ✔ | blue |
| MagicData-RAMC | 2022 | 180 | 663 | 1 | 4 | 4 | 1 | 15 | | ✔ | ✔ | red |
| Kathbath | 2022 | 2k | 1k | 12 | 2 | 2 | 1 | 3 | | ✔ | ✔ | blue |
| Hebrew Kan | 2022 | 9 | - | 1 | 1 | 1 | 1 | 3 | | | | tan |
| Hebrew Coursera | 2022 | 36 | - | 1 | 1 | 1 | 1 | 7 | | | | tan |
| Bloom Speech | 2022 | 428 | - | 56 | 5 | 5 | 1 | 8 | ✔ | ✔ | | blue, red |
| English-Vietnamese | 2022 | 508 | - | 2 | 1 | 1 | 1 | 7 | | | ✔ | red |
| Earnings-22 | 2022 | 119 | 125 | 1 | 1 | 1 | 3 | 2 | ✔ | | ✔ | tan |
| YODAS | 2023 | 370k | - | 149 | 3 | 3 | 1 | 1 | ✔ | ✔ | | blue |
| AFRISPEECH-200 | 2023 | 200 | 2k | 20 | 14 | 14 | 1 | 6 | ✔ | ✔ | ✔ | red |
| Aalto Finnish Parl. | 2023 | 3k | 449 | 1 | 1 | 1 | 1 | 2 | | ✔ | | tan |
| ReazonSpeech | 2023 | 35k | - | 1 | 2 | 2 | 1 | 1 | | | ✔ | blue |
| EdAcc | 2023 | 40 | 120 | 1 | 1 | 1 | 1 | 8 | | ✔ | | blue |
| RixVox | 2023 | 5k | - | 1 | 1 | 1 | 1 | 2 | | | | blue |
| Japanese Anime Speech | 2023 | 110 | - | 1 | 1 | 1 | 1 | 7 | | | | blue |
| Snow Mountain | 2023 | 273 | 11 | 14 | 2 | 2 | 1 | 1 | ✔ | | ✔ | blue |
| Samromur Milljon | 2023 | 967 | 17k | 1 | 1 | 1 | 1 | 5 | | ✔ | | blue |
| Bud500 | 2024 | 500 | - | 1 | 1 | 1 | 2 | 4 | | | | blue, red |
| VibraVox | 2024 | 18 | 200 | 1 | 1 | 1 | 1 | 1 | | ✔ | | blue |
| M2ASR | Mult. | 448 | 655 | 4 | 3 | 3 | 1 | 9 | | ✔ | | tan |

Table 7: **Video collections and properties**. Collection properties include numbers of hours of video, datasets, creator institutions, countries of creator institutions, and data sources. The USE column indicates whether a collection includes data freely usable even for commercial purposes (🔵), data usable only for noncommercial purposes or academic research (🔴) and data whose license status is not specified precisely enough to allow us to determine commercial use permissions (🟡). Note that each collection may have different datasets with one, two, or all three of these statuses. Finally, the AVAIL column indicates whether a dataset is available online (✔) or has been taken down, usually for legal reasons (✗). Datasets are sorted chronologically to highlight trends over time.

| COLLECTION | YEAR | PROPERTY COUNTS | | | | | PERMISSIONS | |
| | | HOURS | DATASETS | COUNTRIES | CREATORS | SOURCES | USE | AVAIL |
| --- | --- | --- | --- | --- | --- | --- | --- | --- |
| HOLLYWOOD2 | 2009 | 20 | 1 | 1 | 1 | 1 | 🟡 | ✔ |
| Collective | 2009 | - | 1 | 1 | 1 | 1 | 🟡 | ✔ |
| HMDB | 2011 | 7k | 1 | 2 | 3 | 5 | 🔵 | ✔ |
| UCF101 | 2012 | 26 | 1 | 1 | 1 | 1 | 🟡 | ✔ |
| YouCook | 2013 | 1k | 1 | 1 | 1 | 1 | 🟡 | ✔ |
| 50 Salads | 2013 | 40 | 1 | 1 | 1 | 1 | 🔴 | ✔ |
| StoryGraphs | 2014 | 7 | 1 | 1 | 1 | 1 | 🟡 | ✔ |
| Hollywood Ext. | 2014 | 9 | 1 | 1 | 1 | 1 | 🔵 | ✔ |
| Breakfast | 2014 | 77 | 1 | 2 | 2 | 1 | 🔵 | ✔ |
| Sports-1M | 2014 | 106k | 1 | 1 | 1 | 1 | 🔵 | ✔ |
| THUMOS | 2014 | 254 | 1 | 2 | 4 | 1 | 🟡 | ✔ |
| VideoStory | 2014 | 743 | 1 | 1 | 1 | 1 | 🟡 | ✔ |
| SumMe | 2014 | 1 | 1 | 2 | 3 | 1 | 🟡 | ✔ |
| TVSum | 2015 | 4 | 1 | 1 | 1 | 1 | 🔵 | ✔ |
| Volleyball | 2015 | - | 1 | 1 | 1 | 1 | 🟡 | ✔ |
| ActivityNet | 2015 | 849 | 1 | 2 | 2 | 1 | 🔵 | ✔ |
| MovieQA | 2015 | 381 | 1 | 3 | 3 | 1 | 🟡 | ✗ |
| Mars | 2016 | - | 1 | 1 | 4 | 1 | 🟡 | ✔ |
| NTU RGB+D | 2016 | 74 | 1 | 1 | 1 | 1 | 🟡 | ✔ |
| MSR-VTT | 2016 | 41 | 1 | 1 | 1 | 1 | 🟡 | ✔ |
| Charades | 2016 | 82 | 1 | 2 | 4 | 1 | 🟡 | ✔ |
| VTW | 2016 | 213 | 1 | 2 | 2 | 1 | 🟡 | ✔ |
| Youtube-8M | 2016 | 350k | 1 | 1 | 1 | 1 | 🟡 | ✔ |
| Narrated Instr. Vid. | 2016 | 7 | 1 | 2 | 4 | 1 | 🔵 | ✔ |
| TGIF | 2016 | 86 | 1 | 1 | 3 | 1 | 🟡 | ✔ |
| MultiTHUMOS | 2017 | 30 | 1 | 2 | 3 | 1 | 🔵 | ✔ |
| ImageNet-Vid | 2017 | 9 | 1 | 1 | 1 | 1 | 🔴 | ✔ |
| PKU-MMD | 2017 | 50 | 1 | 1 | 2 | 1 | 🟡 | ✔ |
| 20BN-SOMETHING | 2017 | 121 | 1 | 1 | 1 | 1 | 🟡 | ✔ |
| YouCook2 | 2017 | 176 | 1 | 1 | 2 | 1 | 🔵 | ✔ |
| VoxCeleb | 2017 | 2k | 1 | 2 | 1 | 1 | 🟡 | ✔ |
| Davis | 2017 | - | 1 | 1 | 2 | 1 | 🟡 | ✔ |
| QFVS | 2017 | 20 | 1 | 1 | 2 | 1 | 🟡 | ✔ |
| DiDeMo | 2018 | 275 | 1 | 1 | 1 | 1 | 🔵 | ✔ |
| SOA | 2018 | 2k | 1 | 1 | 1 | 1 | 🟡 | ✔ |
| Charades-Ego | 2018 | 69 | 1 | 1 | 1 | 1 | 🟡 | ✔ |
| EPIC-KITCHENS | 2018 | 100 | 1 | 3 | 3 | 1 | 🔴 | ✔ |
| MovieGraphs | 2018 | 94 | 1 | 1 | 3 | 1 | 🟡 | ✗ |
| How2 | 2018 | 2k | 1 | 1 | 1 | 1 | 🔴 | ✔ |

Table 7: **Video collections and properties**.

| COLLECTION | YEAR | PROPERTY COUNTS | | | | | PERMISSIONS | |
|---|---|---|---|---|---|---|---|---|
| | | Hours | Datasets | Countries | Creators | Sources | Use | Avail |
| VLOG | 2018 | 336 | 1 | 1 | 1 | 1 | 🟡 | ✔ |
| VaTeX | 2019 | 115 | 1 | 2 | 2 | 1 | 🔵 | ✔ |
| 20BN-jester | 2019 | 13 | 1 | 1 | 1 | 1 | 🟡 | ✔ |
| HowTo100M | 2019 | 134k | 1 | 2 | 4 | 1 | 🟡 | ✔ |
| COIN | 2019 | 476 | 1 | 1 | 2 | 1 | 🟡 | ✔ |
| MMAct | 2019 | 100 | 1 | 2 | 2 | 1 | 🟡 | ✔ |
| HACS | 2019 | 833 | 1 | 1 | 3 | 1 | 🟡 | ✔ |
| CrossTask | 2019 | 376 | 1 | 4 | 5 | 1 | 🟡 | ✔ |
| Moments in Time | 2019 | 833 | 1 | 1 | 1 | 11 | 🟡 | ✔ |
| TRECVid | 2019 | 1k | 1 | 1 | 1 | 2 | 🔴 | ✔ |
| MSA | 2019 | 516 | 1 | 2 | 2 | 1 | 🟡 | ✔ |
| Toyota Smarthome | 2019 | 269 | 1 | 1 | 1 | 1 | 🟡 | ✔ |
| TITAN | 2020 | 3 | 1 | 1 | 1 | 1 | 🔴 | ✔ |
| VIOLIN | 2020 | 582 | 1 | 1 | 1 | 1 | 🟡 | ✔ |
| RareAct | 2020 | 21 | 1 | 3 | 5 | 1 | 🟡 | ✔ |
| TinyVIRAT | 2020 | 11 | 1 | 1 | 1 | 1 | 🟡 | ✔ |
| 100DOH | 2020 | 5k | 1 | 1 | 2 | 1 | 🟡 | ✔ |
| Oops! | 2020 | 50 | 1 | 1 | 1 | 1 | 🔴 | ✔ |
| OmniSource-Web | 2020 | 13k | 1 | 1 | 1 | 3 | 🔵 | ✔ |
| Condensed Movies | 2020 | 1k | 1 | 1 | 1 | 1 | 🔵 | ✔ |
| MovieScenes | 2020 | 250 | 1 | 2 | 2 | 1 | 🟡 | ✔ |
| EEV | 2020 | 370 | 1 | 1 | 2 | 1 | 🔵 | ✔ |
| Movie-Net | 2020 | 3k | 1 | 1 | 1 | 1 | 🟡 | ✔ |
| FineGym | 2020 | 708 | 1 | 1 | 1 | 1 | 🔴 | ✔ |
| HAA500 | 2020 | 5 | 1 | 2 | 4 | 1 | 🟡 | ✔ |
| LEMMA | 2020 | 11 | 1 | 1 | 1 | 2 | 🟡 | ✔ |
| HVU | 2020 | 96k | 1 | 3 | 5 | 1 | 🟡 | ✔ |
| Apes | 2021 | 36 | 1 | 3 | 3 | 1 | 🟡 | ✔ |
| WebVid | 2021 | 13k | 1 | 2 | 2 | 1 | 🟡 | ✗ |
| VideoLT | 2021 | 14k | 1 | 2 | 4 | 1 | 🔴 | ✔ |
| HOMAGE | 2021 | 30 | 1 | 1 | 2 | 1 | 🟡 | ✔ |
| UAV-Human | 2021 | 18 | 1 | 2 | 2 | 1 | 🟡 | ✔ |
| HD-VILA-100M | 2021 | 372 | 1 | 1 | 1 | 1 | 🟡 | ✔ |
| M-MiT | 2021 | 833 | 1 | 1 | 1 | 2 | 🟡 | ✔ |
| Mimetics | 2021 | 1 | 1 | 1 | 1 | 1 | 🟡 | ✔ |
| Spoken Moments | 2021 | 417 | 1 | 1 | 3 | 11 | 🟡 | ✔ |
| QuerYD | 2021 | 207 | 1 | 1 | 1 | 2 | 🟡 | ✔ |
| MAD | 2022 | 1k | 1 | 1 | 1 | 1 | 🟡 | ✔ |
| FERV39k | 2022 | 16 | 1 | 1 | 1 | 1 | 🔴 | ✔ |
| CDAD | 2022 | 215 | 1 | 1 | 2 | 1 | 🟡 | ✔ |
| MVBench | 2023 | - | 1 | 1 | 6 | 12 | 🔵 | ✔ |
| VidProm | 2024 | 240k | 1 | 2 | 2 | 5 | 🔴 | ✔ |
| ShareGPT4Video | 2024 | 3k | 1 | 1 | 4 | 5 | 🔴 | ✔ |
| OpenVid-1M | 2024 | 52k | 1 | 1 | 3 | 5 | 🔵 | ✔ |
| FineVideo | 2024 | 3k | 1 | 1 | 1 | 1 | 🔵 | ✔ |
| Disney Vid. Gen. | 2024 | 7 | 1 | 1 | - | 2 | 🔵 | ✔ |

Table 7: **Video collections and properties**.

| COLLECTION | YEAR | PROPERTY COUNTS | | | | | PERMISSIONS | |
| | | Hours | Datasets | Countries | Creators | Sources | Use | Avail |
|---|---|---|---|---|---|---|---|---|
| Kinetics | Mult. | 4k | 3 | 1 | 1 | 2 | ● | ✔ |
| Ego4D | Mult. | 5k | 2 | 1 | 2 | 1 | ●● | ✔ |
| MPII | Mult. | 110 | 3 | 1 | 2 | 2 | ● | ✔ |
| Project-Aria | Mult. | 1k | 2 | 1 | 1 | 1 | ● | ✔ |
| Ava | Mult. | 146 | 2 | 1 | 1 | 2 | ● | ✔ |
| LSMDC | Mult. | 316 | 2 | 4 | 10 | 1 | ●● | ✔ |

## E  CONTRIBUTIONS

Here we break down contributions to this work. Contributors are listed alphabetically, except for team leads who are placed first.

- **Text Datasets**   Shayne Longpre (lead), Jad Kabbara (lead), Ahmad Anis, Deividas Mataciunas, Diganta Misra, Emad Alghamdi, Enrico Shippole, Jianguo Zhang, Kun Qian, Lester Miranda, Manan Dey, Minnie Liang, Mohammed Hamdy, Nayan Saxena, Niklas Muennighoff, Naana Obeng-Marnu, Robert Mahari, Seonghyeon Ye, Seungone Kim, Shayne Longpre, Shrestha Mohanty, Vipul Gupta, Vivek Sharma, Vu Minh Chien, William Brannon, Xuhui Zhou, Yizhi Li, An Dinh, Caroline Chitongo, Christopher Klamm, Da Yin, Damien Sileo, Ariel Lee

- **Reviewing Text Dataset Metadata**   Jad Kabbara (lead), Shayne Longpre (lead), Robert Mahari, Damien Sileo, Niklas Muennighoff, William Brannon,

- **Data Explorer Features**   Shayne Longpre (lead), Christopher Klamm, Vu Minh Chien,

- **Speech Datasets**  Nikhil Singh (lead), Manuel Cherep (lead), An Dinh, Minnie Liang, Shrestha Mohanty

- **Video Datasets**   Kush Tiwary (lead), Joanna Materzynska (lead), Vivek Sharma, Shayne Longpre, Robert Mahari, Jad Kabbara, William Brannon, Tobin South, Shrestha Mohanty, Nikhil Singh, Manuel Cherep

- **Data Analysis**   Shayne Longpre (lead), Nikhil Singh (lead), Manuel Cherep (lead), Kush Tiwary (lead), Joanna Materzynska (lead), Naana Obeng-Marnu (lead), William Brannon (lead),

- **Writing**   Shayne Longpre (lead), Jad Kabbara (lead), Nikhil Singh, Manuel Cherep, Kush Tiwary, Joanna Materzynska, Robert Mahari

- **Legal Analysis**   Robert Mahari (lead), Luis Villa

- **Visualizations & Visual Data Analysis**   Nikhil Singh (lead), Manuel Cherep (lead), Kush Tiwary (lead), Joanna Materzynska (lead), Naana Obeng-Marnu (lead), William Brannon (lead), Shayne Longpre (lead), Ariel Lee, Hamidah Oderinwale, Campbell Lund

- **Senior Advisors**   Stella Biderman, Sara Hooker, Jad Kabbara, Sandy Pentland, Luis Villa, Caiming Xiong

## F  ATTRIBUTION CARD

Here we provide detailed information about the licenses of each data collection and its constituent datasets, and cite all of the papers (455 in all) which introduced datasets we consider. Text datasets are laid out in Table 8, audio datasets in Table 9, and video datasets in Table 10. Because of the large number of references, we include a second bibliography after the tables (named 'Attribution Card References'), with numbered citations in this section referring to that second bibliography.

Table 8: **References and licenses for alignment-tuning (text)** dataset collections presented in this paper. Collections containing material under more than three distinct licenses are marked as having "Various" licenses, and we refer readers to our raw data for the full details. Datasets are sorted alphabetically for ease of dataset lookup.

| Collection | Licenses | Cite |
|---|---|---|
| 10k Prompt Ranked | Unspecified | – |
| AgentInstruct | Unspecified, CC BY 4.0, MIT License | Shridhar et al. (2021); Yao et al. (2023); Liu et al. (2023c); Zeng et al. (2023); Deng et al. (2023) |
| Aya | Apache License 2.0 | Singh et al. (2024b) |
| Bactrian-X | CC BY-SA 3.0, CC BY-NC 4.0 | Li et al. (2023a) |
| COBRA Frames | BigScience OpenRAIL-M | Zhou et al. (2023b) |
| COIG | Various | Zhang et al. (2023b); Bai et al. (2024a) |
| Capybara | Various | – |
| ChatDoctor | Unspecified | Li et al. (2023d) |
| ChatbotArena | CC BY 4.0, CC BY-NC 4.0 | Zheng et al. (2023) |
| Cidar | CC BY-NC 4.0 | Alyafeai et al. (2024) |
| CollectiveCognition | MIT License | – |
| Conifer | Apache License 2.0 | Sun et al. (2024) |
| Deita 10K | Apache License 2.0, CC BY-NC 4.0 | Liu et al. (2024b) |

Table 8: **References and licenses for alignment-tuning (text)** dataset collections presented in this paper. Collections containing material under more than three distinct licenses are marked as having "Various" licenses, and we refer readers to our raw data for the full details. Datasets are sorted alphabetically for ease of dataset lookup.

| Collection | Licenses | Cite |
|---|---|---|
| DialogStudio | Various | Chen et al. (2021a); Wei et al. (2018); Lin et al. (2021b); Chawla et al. (2021); He et al. (2018); Mrksic et al. (2017); Qian et al. (2022); Liu et al. (2021); El Asri et al. (2017); Quan et al. (2019); Chen et al. (2019; 2022b); Eric & Manning (2017); Zang et al. (2020); Shalyminov et al. (2019); Martin et al. (2020); Peskov et al. (2019); Eric et al. (2019); Moon et al. (2019); Rastogi et al. (2020); Mosig et al. (2020); Chiu et al. (2022); Shah et al. (2018); Byrne et al. (2019); Mrkšić & Vulić (2018); Shang et al. (2018); Rameshkumar & Bailey (2020); Fabbri et al. (2021); Chen et al. (2021c); Mukherjee et al. (2022); Shang et al. (2018); Zhu et al. (2021); Zhong et al. (2021); Gliwa et al. (2019); Chen et al. (2022a); Feigenblat et al. (2021); Li et al. (2019c); Dinan et al. (2019a); Rashkin et al. (2019); Bai et al. (2022); Chen et al. (2023a); Kim et al. (2022); Myers et al. (2020); Reddy et al. (2019); Yu et al. (2019a); Talmor & Berant (2018); Nan et al. (2021; 2022); Gu et al. (2021); Chen et al. (2020b); Gupta et al. (2018a); Li et al. (2021a); Talmor et al. (2021); Yu et al. (2019c); Iyyer et al. (2017); Yu et al. (2019b); Parikh et al. (2020); Yih et al. (2016); Zhong et al. (2017); Pasupat & Liang (2015); Komeili et al. (2022); Dinan et al. (2019b); Hemphill et al. (1990); Casanueva et al. (2020); Zhang et al. (2022b); Larson et al. (2019); Rastogi et al. (2020); Liu et al. (2019; 2013); Coope et al. (2020); Coucke et al. (2018); Gupta et al. (2018b) |

Continued on next page

Table 8: **References and licenses for alignment-tuning (text)** dataset collections presented in this paper. Collections containing material under more than three distinct licenses are marked as having "Various" licenses, and we refer readers to our raw data for the full details. Datasets are sorted alphabetically for ease of dataset lookup.

| Collection | Licenses | Cite |
|---|---|---|
| Dynosaur | Various | Adlakha et al. (2022); Agarwal et al. (2021); Akyürek et al. (2022); Amini et al. (2019); Ardanuy et al. (2020); Austin et al. (2021); Azerbayev et al. (2023); Bai et al. (2022); Bajaj et al. (2018); Balakrishnan et al. (2019); Bartolo et al. (2020); Bisk et al. (2019); Boratko et al. (2020); Botha et al. (2018); Boudin & Gallina (2021); Bravo et al. (2015); Brown et al. (2020b); Byrne et al. (2019); Cao & Wang (2021); Cao et al. (2022); Casanueva et al. (2020); Cetoli et al. (2019); Chalkidis et al. (2019b;a; 2021); Chan et al. (2022); Chapuis et al. (2021); Chen et al. (2020a); Cheng et al. (2022); Chouldechova (2017); Christmann et al. (2019); Clark et al. (2019; 2018); Cobbe et al. (2021); Coucke et al. (2018); Dankers et al. (2022); Dasigi et al. (2019); Devaraj et al. (2021); DeYoung et al. (2021); Diggelmann et al. (2021); Emelin et al. (2020); Fabbri et al. (2019); Faruqui & Das (2018); Feng et al. (2021); Gallina et al. (2019); Ganesan et al. (2010); Gazzola et al. (2019); George & Mamidi (2019); Geva et al. (2019); Gliwa et al. (2019); Gorrell et al. (2018); Gu et al. (2022); Gupta et al. (2021); Ha & Eck (2017); Haagsma et al. (2020); Hazoom et al. (2021); Henderson et al. (2022); Hendrycks et al. (2021); Huang et al. (2019; 2021); Huang (2022); Irwin et al. (2020); Ivgi et al. (2022); Iyer et al. (2017); Jiang et al. (2020; 2021); Jin et al. (2019); Joshi et al. (2017); Juraska et al. (2019); Jurczyk et al. (2016); Kanade et al. (2020); Kaushik et al. (2020); Khot et al. (2020; 2018); Kim et al. (2018); Kornilova & Eidelman (2019); Kury et al. (2020); Lai et al. (2017); Lake & Baroni (2018); Lebret et al. (2016); Lewis et al. (2017); Li et al. (2022; 2019a); Lin et al. (2020a) |

Table 8: **References and licenses for alignment-tuning (text)** dataset collections presented in this paper. Collections containing material under more than three distinct licenses are marked as having "Various" licenses, and we refer readers to our raw data for the full details. Datasets are sorted alphabetically for ease of dataset lookup.

| Collection | Licenses | Cite |
|---|---|---|
| Dynosaur (cont'd) | Various | Lin et al. (2020b; 2019; 2022); Ling et al. (2017); Liu et al. (2019); Louis et al. (2020); Lowe et al. (2016); Malo et al. (2013); Martin et al. (2018); Merity et al. (2016); Mihaylov et al. (2018); Mishra et al. (2023); Moniz & Torgo (2018); Mostafazadeh et al. (2020); Nan et al. (2021); Narayan et al. (2018); Nguyen et al. (2021); Nie et al. (2020); Novikova et al. (2017); Paik et al. (2021); Pakhomov et al. (2010); Pang & Lee (2005); Pavlichenko et al. (2021); Pedersen et al. (2007); Perez-Beltrachini et al. (2019); Petroni et al. (2019); Pham et al. (2023); Rajani et al. (2019); Rajpurkar et al. (2016b); Rameshkumar & Bailey (2020); Rashkin et al. (2019); Rastogi et al. (2020); Royer et al. (2018); Rush et al. (2015); Rust et al. (2023); Saeidi et al. (2018); Saha et al. (2018); Sakaguchi et al. (2019); Sanh et al. (2022); Sap et al. (2019); Schulz et al. (2020); See et al. (2017); Sharma et al. (2019); Shriberg et al. (1998); Sileo & Moens (2023); Soleimani et al. (2021); Stolcke et al. (2000); Tafjord et al. (2019; 2018); Talmor et al. (2019); Tandon et al. (2019); Tang et al. (2020); Thawani et al. (2021); Thorne et al. (2018); Tyleček & Šára (2013); Ullrich et al. (2023); Ushio et al. (2023); Wang et al. (2022a; 2020c; 2019; 2023a); Warstadt et al. (2023); Welbl et al. (2018; 2017); Weller et al. (2020); Weston et al. (2015); Williams et al. (2020); Wu et al. (2018); Xiong et al. (2019a); Yang et al. (2018); Yu et al. (2019b); Zellers et al. (2019); Zhang et al. (2016; 2023c; 2019); Zhou et al. (2019; 2023a); Zhu et al. (2022) |
| EverythingLM | MIT License | – |
| ExpertQA | MIT License | Malaviya et al. (2024) |
| Feedback Coll. | MIT License | Kim et al. (2024) |

Continued on next page

Table 8: **References and licenses for alignment-tuning (text)** dataset collections presented in this paper. Collections containing material under more than three distinct licenses are marked as having "Various" licenses, and we refer readers to our raw data for the full details. Datasets are sorted alphabetically for ease of dataset lookup.

| Collection | Licenses | Cite |
|---|---|---|
| Glaive Code Asst. | Apache License 2.0 | – |
| Gretel Text-to-SQL | Apache License 2.0 | – |
| HelpSteer | CC BY 4.0 | Wang et al. (2023b) |
| Indic-Instr. | Various | Gala et al. (2024) |
| InstAr | Various | Hu et al. (2020); Einea et al. (2019); Mozannar et al. (2019); Pratapa et al. (2022); Chouikhi et al. (2024); Abbas et al. (2011); Abdelghany et al. (2020); Orabi et al. (2020); ElSahar & El-Beltagy (2015); Elnagar & Einea (2016); Pieri et al. (2024); Alghamdi et al. (2022); Abdallah et al. (2024); Biltawi et al. (2020); Aloui et al. (2024); El-khair (2016) |
| KIWI | CC BY-SA 4.0 | Xu et al. (2024a) |
| LMSYS-Chat-1M | LMSYS-Chat-1M Dataset License, Anthropic, Llama 2 | Zheng et al. (2024a) |
| Llama2-MedTuned-Instr. | CC BY-NC 4.0 | Rohanian et al. (2023) |
| LongAlign-10k | Anthropic, Apache License 2.0 | Bai et al. (2024b) |
| Magpie-Pro | Meta Llama3 Community License | Xu et al. (2024b) |
| MathDial | CC BY-SA 4.0, MIT License | Macina et al. (2023) |
| MathInstr. | MIT License | Yue et al. (2023) |
| MedInstr. | Unspecified | Zhang et al. (2024) |
| Medical Meadow | Various | Han et al. (2023); Wang et al. (2020b); Jin et al. (2020); Savery et al. (2020) |
| MegaWika | CC BY-SA 4.0 | Barham et al. (2023) |
| MetaMathQA | MIT License | Yu et al. (2023) |
| Nectar | Various | – |
| No Robots | CC BY-NC 4.0 | – |
| Open Asst. v2 | Apache License 2.0 | Köpf et al. (2023) |
| Open-Platypus | Various | Sawada et al. (2023); Dettmers et al. (2023); Lightman et al. (2023); Yu et al. (2020); Wang et al. (2024); Lu et al. (2022); Chen et al. (2023b) |
| OpenGPT Healthcare | Unspecified, OGL 3.0 | – |
| OpenMathInstr.-1 | Custom, MIT License, Apache License 2.0 | Toshniwal et al. (2024) |
| Orca-Math | Various | Mitra et al. (2024) |
| PII-Masking-200k | Non Commercial | – |
| PMC-LLaMA Instr. | Unspecified, Apache License 2.0 | Wu et al. (2023); Jin et al. (2019) |
| Pure-Dove | Apache License 2.0 | – |
| PygmalionAI-PIPPA | Apache License 2.0 | Gosling et al. (2023) |

Table 8: **References and licenses for alignment-tuning (text)** dataset collections presented in this paper. Collections containing material under more than three distinct licenses are marked as having "Various" licenses, and we refer readers to our raw data for the full details. Datasets are sorted alphabetically for ease of dataset lookup.

| Collection | Licenses | Cite |
|---|---|---|
| Reasoning | Apache License 2.0 | – |
| RiddleSense | MIT License | Lin et al. (2021a) |
| SeaBench | Apache License 2.0 | Nguyen et al. (2023) |
| SelFee | MIT License | Ye et al. (2023) |
| Synth.-GSM8K-Refl. | Meta Llama3 Community License | – |
| Thai Gen AI | Various | – |
| UltraChat-200k | CC BY-NC 4.0 | Ding et al. (2023) |
| UltraFeedback Argilla | Various | – |
| WildChat | AI2 ImpACT License - Low Risk | Zhao et al. (2023) |

Table 9: **References and licenses for audio** dataset collections presented in this paper. Collections containing material under more than three distinct licenses are marked as having "Various" licenses, and we refer readers to our raw data for the full details. Datasets are sorted alphabetically for ease of dataset lookup.

| Collection | Licenses | Cite |
|---|---|---|
| 1111 Hours Hindi | Custom | Bhanushali et al. (2022) |
| 120h Spanish Speech | CC0 1.0 | – |
| AFRISPEECH-200 | CC BY-NC-SA 4.0 | Olatunji et al. (2023) |
| AISHELL-1 | Apache 2.0 | Bu et al. (2017) |
| AISHELL-2 | Unspecified | Du et al. (2018) |
| AISHELL-4 | CC BY-SA 4.0 | Fu et al. (2021) |
| ALLSSTAR | CC BY 4.0 | Bradlow (2010) |
| AMI | CC BY 4.0 | Carletta et al. (2006) |
| Aalto Finnish Parl. | Custom | Virkkunen et al. (2022) |
| African Acc. French | Apache 2.0 | – |
| Basq., Cat. and Gal. | CC BY-SA 4.0 | Kjartansson et al. (2020) |
| Bloom Speech | Various | Leong et al. (2022) |
| Bud500 | Apache 2.0, CC BY-NC-SA 4.0 | – |
| CSJ | Custom | Maekawa (2003) |
| CSLU 1.2 | CSLU Agreement | Lander, T (2007) |
| CSLU 22 Langs. | CSLU Agreement | Lander, T (2005) |
| ClarinPL | CC BY 4.0 | Korzinek et al. (2017) |
| CoNASE | Custom | Coats (2019) |
| CoVoST-2 | CC0 1.0 | Wang et al. (2020a) |
| Common Voice 17 | CC0 1.0 | Ardila et al. (2020a) |
| Crowd Sourced Speech | CC BY-SA 4.0 | Kjartansson et al. (2018) |
| Czech Parliament | CC BY 4.0 | Kratochvil et al. (2020) |
| DiDiSpeech | Unspecified | Guo et al. (2021) |
| Earnings-22 | Unspecified | Del Rio et al. (2022) |
| EdAcc | CC BY-SA 4.0 | Sanabria et al. (2023) |
| Eng. Acc. in Brit. Isles | CC BY-SA 4.0 | Demirsahin et al. (2020) |
| English-Vietnamese | CC BY-NC-ND 4.0 | Nguyen et al. (2022) |
| FT SPEECH | Custom | Kirkedal et al. (2020) |
| Fisher | LDC User Agreement | Cieri et al. (2004) |
| Fleurs | CC BY 4.0 | Conneau et al. (2022) |
| GigaSpeech | Apache 2.0 | Chen et al. (2021b) |
| Golos | Custom | Karpov et al. (2021) |
| Hebrew Coursera | Unspecified | – |
| Hebrew Kan | Unspecified | – |
| Highland Puebla Nahuatl | CC BY-NC-SA 3.0 | Shi et al. (2021) |
| JTUBESPEECH | Unspecified | Takamichi et al. (2021) |
| Japanese Anime Speech | CC0 1.0 | – |
| KSC | CC BY 4.0 | Khassanov et al. (2021) |
| Kathbath | CC0 1.0 | Javed et al. (2022) |
| KeSpeech | Custom | Tang et al. (2021) |
| KsponSpeech | Unspecified | Bang et al. (2020) |

Table 9: **References and licenses for audio** dataset collections presented in this paper. Collections containing material under more than three distinct licenses are marked as having "Various" licenses, and we refer readers to our raw data for the full details. Datasets are sorted alphabetically for ease of dataset lookup.

| Collection | Licenses | Cite |
| --- | --- | --- |
| LJSpeech | Public Domain | Ito & Johnson (2017) |
| LaboroTVSpeech | Custom | Ando & Fujihara (2021) |
| LibriSpeech | CC BY 4.0 | Panayotov et al. (2015) |
| M-AILABS | Custom | Solak (2024) |
| M2ASR | Unspecified | Shi et al. (2017); Mamtimin et al. (2023); Zhi et al. (2021); Li et al. (2017) |
| MAGICDATA | CC BY-NC-ND 4.0 | – |
| MASC | CC BY 4.0 | Al-Fetyani et al. (2023) |
| MaSS | Unspecified | Boito et al. (2020) |
| MagicData-RAMC | CC BY-NC-ND 4.0 | Yang et al. (2022) |
| MediaSpeech | CC BY 4.0 | Kolobov et al. (2021) |
| Minds14 | CC BY 4.0 | Gerz et al. (2021) |
| MuST-C | CC BY-NC-ND 4.0 | Di Gangi et al. (2019) |
| Multiling. LibriSpeech | CC BY 4.0 | Pratap et al. (2020) |
| Multiling. TEDx | CC BY-NC-ND 4.0 | Salesky et al. (2021) |
| NST Danish | CC0 1.0 | – |
| NST Norwegian | CC0 1.0 | – |
| NST Swedish | CC0 1.0 | – |
| Nigerian English | CC BY-SA 4.0 | – |
| Norwegian Parl. | CC0 1.0 | Solberg & Ortiz (2022) |
| Norwegian Parl. Speech | CC0 1.0 | Solberg & Ortiz (2022) |
| OLKAVS | Custom | Park et al. (2023) |
| OpenSTT | CC BY-NC 4.0 | Andrusenko et al. (2020) |
| People's Speech | Various | Galvez et al. (2021) |
| QASR | Unspecified | Mubarak et al. (2021) |
| RTVE | Custom | – |
| ReazonSpeech | CDLA-Sharing-1.0 | Yin et al. (2023) |
| Regional Af. Am. Lang. | CC BY-NC-SA 4.0 | – |
| RixVox | CC BY 4.0 | – |
| SDS-200 | Custom | Plüss et al. (2022) |
| SPGISpeech | Custom | O'Neill et al. (2021) |
| Samromur | CC BY 4.0 | Mollberg et al. (2020) |
| Samromur Children | CC BY 4.0 | Hernandez Mena et al. (2022) |
| Samromur Milljon | CC BY 4.0 | – |
| Shrutilipi | CC0 1.0 | Bhogale et al. (2022) |
| Snow Mountain | CC BY-SA 4.0 | Raju et al. (2023) |
| Spoken Wikipedia | CC BY-SA 4.0 | Baumann et al. (2019) |
| Switchboard | LDC User Agreement | Godfrey et al. (1992) |
| TED-LIUM3 | CC BY-NC-ND 3.0 | Hernandez et al. (2018) |
| THCHS-30 | Apache 2.0 | Wang & Zhang (2015) |
| THUYG-20 | Apache 2.0 | Rozi et al. (2015) |

Table 9: **References and licenses for audio** dataset collections presented in this paper. Collections containing material under more than three distinct licenses are marked as having "Various" licenses, and we refer readers to our raw data for the full details. Datasets are sorted alphabetically for ease of dataset lookup.

| Collection | Licenses | Cite |
|---|---|---|
| TIMIT | LDC User Agreement | Garofolo, John S. et al. (1993) |
| VCTK | CC BY 4.0 | – |
| VibraVox | CC BY 4.0 | – |
| VoxPopuli | CC0 1.0 | Wang et al. (2021) |
| Vystadial | CC BY-SA 3.0 | Korvas et al. (2014) |
| WenetSpeech | CC BY 4.0 | Zhang et al. (2022a) |
| West Afr. Radio | CC BY-SA 4.0 | Doumbouya et al. (2021) |
| West Afr. Virt. Asst. | CC BY-SA 4.0 | Doumbouya et al. (2021) |
| YODAS | CC BY 3.0 | Li et al. (2023c) |
| Zeroth-Korean | CC BY 4.0 | – |
| aidatatang | CC BY-NC-ND 4.0 | – |

Table 10: **References and licenses for video** dataset collections presented in this paper. Collections containing material under more than three distinct licenses are marked as having "Various" licenses, and we refer readers to our raw data for the full details. Datasets are sorted alphabetically for ease of dataset lookup.

| Collection | Licenses | Cite |
|---|---|---|
| 100DOH | Custom | Shan et al. (2020) |
| 20BN-SOMETHING | Custom | Goyal et al. (2017) |
| 20BN-jester | Custom | Materzynska et al. (2019) |
| 50 Salads | CC BY-NC-SA 4.0 | Stein & McKenna (2013) |
| ActivityNet | MIT License | Heilbron et al. (2015) |
| Apes | Unspecified | Alcazar et al. (2021) |
| Ava | CC BY 4.0 | Roth et al. (2019); Gu et al. (2018) |
| Breakfast | CC BY 4.0 | Kuehne et al. (2014) |
| CDAD | Unspecified | Xiang et al. (2022) |
| COIN | Custom | Tang et al. (2019) |
| Charades | Custom | Sigurdsson et al. (2016b) |
| Charades-Ego | Custom | Sigurdsson et al. (2018) |
| Collective | Unspecified | Wongun Choi et al. (2009) |
| Condensed Movies | CC BY 4.0 | Bain et al. (2020) |
| CrossTask | Unspecified | Zhukov et al. (2019) |
| Davis | Custom | Perazzi et al. (2016) |
| DiDeMo | BSD 2-Clause License | Hendricks et al. (2018) |
| Disney Vid. Gen. | Apache 2.0 | – |
| EEV | CC BY 4.0 | Sun et al. (2021) |
| EPIC-KITCHENS | CC BY-NC 4.0 | Damen et al. (2018) |
| Ego4D | Custom, MIT License | Grauman et al. (2022) |
| FERV39k | CC BY-NC 4.0 | Wang et al. (2022b) |
| FineGym | CC BY-NC 4.0 | Shao et al. (2020) |
| FineVideo | CC BY 4.0 | – |
| HAA500 | Unspecified | Chung et al. (2021) |
| HACS | Custom | Zhao et al. (2019) |
| HD-VILA-100M | Custom | Xue et al. (2022) |
| HMDB | CC BY 4.0 | Kuehne et al. (2011) |
| HOLLYWOOD2 | Unspecified | Marszalek et al. (2009) |
| HOMAGE | Unspecified | Rai et al. (2021) |
| HVU | Custom | Diba et al. (2020) |
| Hollywood Ext. | MIT License | Bojanowski et al. (2014) |
| How2 | Various | Sanabria et al. (2018) |
| HowTo100M | Unspecified | Miech et al. (2019) |
| ImageNet-Vid | CC BY-NC 4.0 | Russakovsky et al. (2015) |
| Kinetics | Unspecified | Kay et al. (2017); Carreira et al. (2018); Smaira et al. (2020) |
| LEMMA | Unspecified | Jia et al. (2020) |
| LSMDC | Custom, MIT License | Rohrbach et al. (2016a); Sharma et al. (2020) |
| M-MiT | Unspecified | Monfort et al. (2021b) |

Continued on next page

Table 10: **References and licenses for video** dataset collections presented in this paper. Collections containing material under more than three distinct licenses are marked as having "Various" licenses, and we refer readers to our raw data for the full details. Datasets are sorted alphabetically for ease of dataset lookup.

| Collection | Licenses | Cite |
|---|---|---|
| MAD | Custom | Soldan et al. (2022) |
| MMAct | Custom | Kong et al. (2019) |
| MPII | Unspecified, Custom | Rohrbach et al. (2016b; 2015) |
| MSA | Unspecified | Xiong et al. (2019b) |
| MSR-VTT | Unspecified | Xu et al. (2016) |
| MVBench | MIT License | Li et al. (2024) |
| Mars | Unspecified | Zheng et al. (2016) |
| Mimetics | Unspecified | Weinzaepfel & Rogez (2021) |
| Moments in Time | Custom | Monfort et al. (2019b) |
| Movie-Net | Unspecified | Huang et al. (2020) |
| MovieGraphs | Custom | Vicol et al. (2018) |
| MovieQA | Unspecified | Tapaswi et al. (2016) |
| MovieScenes | Unspecified | Rao et al. (2020) |
| MultiTHUMOS | CC BY 4.0 | Yeung et al. (2017) |
| NTU RGB+D | Custom | Shahroudy et al. (2016) |
| Narrated Instr. Vid. | MIT License | Alayrac et al. (2016) |
| OmniSource-Web | Apache License 2.0 | Duan et al. (2020) |
| Oops! | CC BY-NC-SA 4.0 | Epstein et al. (2020) |
| OpenVid-1M | CC-BY-4.0 | Nan et al. (2024) |
| PKU-MMD | Unspecified | Liu et al. (2017) |
| Project-Aria | Apache License 2.0 | Pan et al. (2023); Lv et al. (2024) |
| QFVS | Unspecified | Sharghi et al. (2017) |
| QuerYD | Unspecified | Oncescu et al. (2021) |
| RareAct | Unspecified | Miech et al. (2020) |
| SOA | Unspecified | Diba et al. (2020) |
| ShareGPT4Video | Attribution-NonCommercial 4.0 International | Chen et al. (2024) |
| Spoken Moments | Custom | Monfort et al. (2021a) |
| Sports-1M | CC BY 3.0 | Karpathy et al. (2014) |
| StoryGraphs | Unspecified | Tapaswi et al. (2014) |
| SumMe | Unspecified | Gygli et al. (2014) |
| TGIF | Custom | Li et al. (2016) |
| THUMOS | Custom | Idrees et al. (2017) |
| TITAN | Non Commercial | Malla et al. (2020) |
| TRECVid | CC BY-NC-SA 4.0 | Awad et al. (2020) |
| TVSum | CC BY 3.0 | Yale Song et al. (2015) |
| TinyVIRAT | Unspecified | Demir et al. (2020) |
| Toyota Smarthome | Custom | Das et al. (2019) |
| UAV-Human | Custom | Li et al. (2021b) |
| UCF101 | Unspecified | Soomro et al. (2012) |
| VIOLIN | Unspecified | Liu et al. (2020) |
| VLOG | Custom | Fouhey et al. (2017) |

Table 10: **References and licenses for video** dataset collections presented in this paper. Collections containing material under more than three distinct licenses are marked as having "Various" licenses, and we refer readers to our raw data for the full details. Datasets are sorted alphabetically for ease of dataset lookup.

| Collection | Licenses | Cite |
|---|---|---|
| VTW | Unspecified | Zeng et al. (2016) |
| VaTeX | CC BY 4.0 | Wang et al. (2020d) |
| VidProm | CC-BY-NC 4.0 | Wang & Yang (2024b) |
| VideoLT | Non Commercial | Zhang et al. (2021) |
| VideoStory | Unspecified | Habibian et al. (2014) |
| Volleyball | Unspecified | Ibrahim et al. (2016) |
| VoxCeleb | Custom | Nagrani et al. (2018) |
| WebVid | Custom | Bain et al. (2022) |
| YouCook | Unspecified | Das et al. (2013) |
| YouCook2 | MIT License | Zhou et al. (2017) |
| Youtube-8M | Unspecified | Abu-El-Haija et al. (2016a) |

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
