# OpenReview forum: "Bridging the Data Provenance Gap Across Text, Speech, and Video"
_ICLR.cc/2025/Conference — ICLR 2025 Poster_

### Official Review · Reviewer_BAwp · 2024-10-30

**Soundness:** 3
**Presentation:** 3
**Contribution:** 3
**Rating:** 8
**Confidence:** 3

**Summary:**

The authors conduct a comprehensive audit across modalities—popular text, speech, and video datasets—from their detailed sourcing trends and use restrictions to their geographical and linguistic representation.

**Strengths:**

1. This is an extremely exciting work for multi-modal researchers, as one of the main bottlenecks is how to get balanced and good quality data from the wild. This work deepens the communities understanding of multi-modal data conditions at present and in the history.
2. The authors conducted rigorous data collection and analysis work, which deserves much credit.
3. The insights provided are valuable and interesting, which may provide great inspirations for people curating multi-modal data from real world.

**Weaknesses:**

1. I feel the data collection specifications are limited. Though the authors has mentioned that the discussion for geographical and linguistic representation is limited, I think more discussion should be casted on the data collection methods, as this is the foundation of all discussions in this work. How to ensure the recall and precision of data collection? The data and its conclusion could easily get biased when missing or mistakenly putting a large dataset into the data pool.

**Questions:**

1. I'm curious that did you perform analysis on the portion of synthetic data and their characteristics? I think this is one of the most concerned topics by far, though it's really hard to correctly collect all the data.

---

> ### Author Response · Authors · 2024-11-21
>
> We are extremely encouraged by Reviewer BAwp’s positive feedback, and recognition that this work addresses central bottlenecks in sourcing quality data in the wild. We also appreciate that the reviewer finds our analysis rigorous and highly relevant to real world practices. The reviewer has suggested that we better justify the data collection procedures that underlie the analysis, and extend our analysis of synthetic data. We propose changes below to address these recommendations, which we will incorporate into the paper.
>
> **How do the data collection & scoping methodologies ensure the analysis is empirically sound?**
>
> This is an important question, and one we agree is at the heart of the analysis. We spent a lot of time discussing and testing strategies for data collection and scoping methodology before collection. We would like to emphasize that by nature, this problem is difficult to solve: under-scoping datasets to just those found in one journal/conference, or even those already in a set of journals, is easier to define but can miss out on massive and important datasets. Many popular datasets are posted on HuggingFace or GitHub, not even with an arxiv paper, far less published to journals/conferences. Accordingly, to capture the broader “ecosystem” (where more impactful research questions can be answered), we realized it is necessary to go beyond this, and that scoping and discovery would need to be extremely rigorous.
>
> We addressed these concerns in the following way:
>
> We structured the search with a dedicated team for each modality led by experts within that domain. Each team relied on (a) their own expertise within that domain, (b) a mix of search tags from HuggingFace Datasets, (c) survey papers, (d) survey repositories, and (e) workshop/conference proceedings. External experts were also reached out to, in each case, to look for gaps in the coverage. Note that where possible, we sought to leverage prior work. Survey papers and repositories were particularly useful, as many are community crowdsourced, and driven by usage more than peer review, like workshops.
>
> Overall, it is virtually impossible to be absolutely certain that the dataset coverage is perfect, but we believe this methodology is the most rigorous attempt to capture the full ecosystem, and we are fairly confident all widely used datasets are captured for each modality, within their predefined scopes. Moreover, as Reviewer Kmty pointed out, this paper is one of the first in this direction and our tools and annotations serve a call to action for the community, future dataset creators and their users.
>
> Further, our goal is to ensure the database can be easily reused or extended, to facilitate continued research, as new datasets are added or scopes expanded. We will revisit this discussion in the methodology to better justify the process and more thoroughly describe the details.
>
> **Have we analyzed the characteristics of the synthetic data?**
>
> Yes—we consolidated a few key points in our general response above, and will add them to the paper.
>
> Thanks again for your positive feedback and comments! Please let us know if you have further clarifications or suggestions.

---

> > ### Comment · Reviewer_BAwp · 2024-11-22
> >
> > Thanks for the response! Almost all my concerns are resolved

---

### Official Review · Reviewer_AQ3r · 2024-11-02

**Soundness:** 4
**Presentation:** 4
**Contribution:** 4
**Rating:** 8
**Confidence:** 3

**Summary:**

This work conducts a large-scale and comprehensive audit of nearly 4,000 text, speech, and video datasets between 1990-2024, covering a wide range of tasks, languages, sources, organizations, and countries. Specifically, the authors analyze data trends for the state of data permissions (licenses and terms), sourcing (the web, human annotation, and synthetic generation), and representation (of tasks, organizations, languages, and countries). The datasets included in the audit include those that are publicly available, widely used for general-purpose model development, and relevant to generative tasks. Overall, the authors analyzed 3.7k text, 95 speech, and 104 video datasets.

The analysis revealed an increasing use of data from web-crawled, social media, and synthetic sources, driven by scaling laws. This was particularly prevalent in speech and video datasets, with YouTube being the most prominent source. The contribution of synthetic text data has also rapidly become significant in recent years. Additionally, the authors found inconsistencies in data source terms and documented license restrictions, with the former being much more restrictive. Further, many data collections do not clarify commercially licensed versus restrictive datasets. Finally, the analysis revealed that while there has been an increase in the languages and countries represented in the datasets, the relative contribution remains Western-centric.

**Strengths:**

* The paper provides novel insights from a large-scale and comprehensive audit of widely used datasets across text, speech, and video modalities.
* The paper is well-written and organized.
* Larger implications of the authors’ findings are discussed, highlighting challenges and suggestions for practitioners.

**Weaknesses:**

* The audit does not include image datasets, which are widely used in multimodal settings.
* The analysis does not include the tasks that the datasets are designed for.
* Some more insights and analysis can be provided for the cause of observed trends- e.g. the sharp rise in encyclopedia-based and internet video-based sources in text and speech datasets after 2018, the drop in the Gini coefficient for geographic representation in video datasets after 2019, etc.

**Questions:**

* Can the source of synthetic datasets (e.g. the models they were generated from) be determined? Are there any challenges in determining such sources?
* Why was the scope of the datasets limited to generative tasks?
* Was there a variation in the tasks represented by the data across the years? It would be interesting to observe emerging trends in AI research over the years (e.g. code generation and other domain-specific tasks).
* Was the correlation between data sources and language/geographical diversity analyzed?

---

> ### Author Response · Authors · 2024-11-21
>
> We would like to thank Reviewer AQ3r for their constructive feedback and their strong support of our submission. We are particularly encouraged that they recognized the novel insights, and larger implications of the findings to practitioners. Their feedback centers around further analysis on dataset task trends, synthetically-generated data, and correlations between observed features. Based on their questions, we have run some numbers on our dataset annotations, and are happy to add these deeper analyses into the paper and appendix.
>
> **Can the source of synthetic datasets (e.g. the models they were generated from) be determined?**
>
> Yes! We’ve consolidated this information in the general response. Based on your comment, we’re happy to clarify this distinction further in the appendix of the camera-ready version.
>
> **Why was the scope of the datasets limited to generative tasks?**
>
> This is a great question. We decided to ground the work in tasks with *relevance* to generation, to find a reasonable bound to the scope and concentrate on generative models rather than purely discriminative tasks. Note that *relevance* to generation includes many tasks that are also discriminative but have been repurposed—e.g. Text classification datasets are widely used for instruction finetuning, once reformatted. For video, classification tasks such as Action Classification are actually considered generation as the video-label pair can be used in a generator to actually train such a video generation model. We provide a detailed note of what types of video tasks datasets are considered in the appendix. . All speech datasets are primarily for speech recognition, which is not always modeled as a generative task. Many can also be used for other tasks, which can be generative or discriminative. We will add further details in each modality’s methodology subsection.
>
> **Was there a variation in the tasks represented by the data across the years? It would be interesting to observe emerging trends in AI research over the years (e.g. code generation and other domain-specific tasks).**
>
> Thanks for raising this point! We agree that task trends are very interesting. While it didn’t make it into the main paper due to page constraints, we actually did collect fine-grained task metadata for every dataset, and report the top-level category distributions for each modality in Figure 7 (Appendix: “Additional Results”).
>
> Prompted by your question, we plotted the increase in tokens and hours for each modality. Some clear trends emerge. For text, we immediately noticed that Code Generation has grown significantly since 2020-2022 whereas many more traditional NLP tasks like Summarization, Multiple Choice Question, and Question Answering have relatively subsided. For Video we notice emerging trends in Video Captioning, Video Q&A and Video Summarization have grown immensely in the 2020s. Additionally, we also see that Video Classification as a task has subsided in popularity, compared to its rise in 2013-2016, as more complex tasks such as Captioning and summarization have seen more success. We’d be happy to add these plots and analysis into the paper and appendix.
>
> **Was the correlation between data sources and language/geographical diversity analyzed?**
>
> This is also a great question. We conducted a preliminary analysis, first mapping countries to continents and consolidating categories to reduce sparsity. We then performed $\chi^2$ tests of independence (with Bonferroni correction across modalities) and Cramer's V effect sizes. We observed statistically significant associations for both video ($\chi^2=70.67, p<.0001, V=.302$) and text ($\chi^2=8014.76, p<0.0001, V=0.258$) with moderate effect sizes, but not speech ($\chi^2=24.33, p=0.434, V=0.248$) due to the prevalence of human and video sourced data across continents.
>
> However, these results should be interpreted with caution due to prevailing sparsity despite our category aggregation. We view this as an important preliminary finding that warrants further investigation, potentially with more coarse-grained analyses better suited to sparse data. A more thorough analysis may reveal ways in which creators from different regions systematically differ in their data sources, adding context to our understanding of how geographic impacts dataset creation.
>
> Finally, we’d like to thank you again for your positive feedback and incisive comments! Please let us know if you have further clarifications or suggestions.

---

> > ### Comment · Reviewer_AQ3r · 2024-11-26
> >
> > Dear authors,
> > Thank you for addressing the questions. The analysis of the source of synthetic datasets is particularly interesting and it would be great to include it in the appendix or elsewhere in the paper.

---

### Official Review · Reviewer_Kmty · 2024-11-04

**Soundness:** 3
**Presentation:** 3
**Contribution:** 2
**Rating:** 5
**Confidence:** 4

**Summary:**

In this paper, the authors present an audit of nearly 4000 public datasets between 1990-2024, spanning 608 languages, 798 sources, 659 organizations, and 67 countries. They further present their analysis of data sources, geographical and multilingual inclusion.

**Strengths:**

The paper is well written and easy to understand. The authors explore an important topic relevant to extremely fast-paced growth of AI models and their adoption. They explore a large-scale collection of datasets across three modalities: text, speech and video. They further present interesting analysis highlighting the community inclusion for dataset creation and the restriction on the usage of the datasets. The study brings out the inequality in geographical representation and the multilingual representation and furthermore the situation has not improved significantly over the last decade on most measures.

**Weaknesses:**

I find the lack of call-to-action from the authors. There are no concreate directions presented in the paper that could help improve the situation. A mere analysis of the landscape is probably not going to be of much help given the pace with which the field is evolving. They have surveyed so many papers which should have nudged them to form an opinion on the next course of action for the community.

Also, since this field is growing rapidly, the analysis presented here would become obsolete very quickly. What might help in the long-term is to build a tool that can automate the analysis presented here with such dedicated effort.

**Questions:**

See my comments in the weaknesses

---

> ### Author Response · Authors · 2024-11-21
>
> We would like to thank Reviewer Kmty for their constructive feedback and thoughtful critiques. We appreciate their recognition of the importance of the findings around geographical representation and multilingual representation. The reviewer asks that we provide clearer calls to action and justify the long term value of the contributions. Below, we address both by pointing out relevant but underemphasized parts of the current paper, and propose changes that (a) center concrete calls to action and (b) discuss the longer term foundation this work provides for future research.
>
> **Ensuring the analysis can keep pace with a rapidly growing field**
>
> We agree it is important to enable continued data measurement for these contributions to remain relevant. However, we also do believe that large-scale retrospective analyses are valuable to inform the community in their own right. There are multiple ways this work is designed to facilitate continuous ecosystem-wide “data measurement”:
> * First, we are open sourcing not just the annotations, but several pieces that helped us scale up the collection and analysis: (a) all annotation procedures/instructions (including the legal analysis, designed by domain experts), (b) the code for aggregating metadata from GitHub, HuggingFace, Papers with Code, Semantic Scholar, and (c) our many contributed taxonomies and mappings for organization geographies, task hierarchies, or use restrictions for ~100+ license types. All of these are organized to enable faster iteration and expedite new analysis. We believe this to be an important contribution to the wider community.
> * Second, we’ve made an effort to identify and retroactively annotate three decades of historical datasets (1990 - 2024). Future analyses can simply import our CSVs and compare their own dataset domains against these or contribute the incremental additions back to our collection.
> * Third, we are continuing to facilitate new contributions, and are working on having it crowdsourced, with internal validation checks so this can scale through community effort going forward while maintaining annotation quality. Our hope and vision is that our work’s contribution is not a fixed snapshot in time, but also a continued research effort that rallies stakeholders from various sectors.
>
> All this is to say, we agree with your point, and are working towards addressing it, as well as providing a solid foundation for new analyses to adopt and build on. There are many pieces to our audit that this part wasn’t as emphasized, but based on your feedback, we will include the points above.

---

> > ### Author Response · Authors · 2024-11-21
> > **(Part 2—Actionable Recommendations)**
> >
> > **Strengthening our call to action**
> >
> > We take this point as recommending we present a call to action in our work and explain how our work will benefit the AI community going forward. We agree with reviewer Kmty about the importance of this topic and strongly believe in the impact of this work. To address this suggestion, we propose to extend the discussion section to outline how our paper provides **actionable recommendations** to the community. Specifically, we will clearly outline the needs and disparities in the community that our work both identifies and suggests paths to mitigating.
> >
> > Broadly speaking, we believe our work provides much-needed transparency to the multimodal data ecosystem, which can mitigate several serious risks and harms. Our work points the way toward changes in current practice that can make this level of transparency the baseline. We expand on these below:
> > * First, our work facilitates future and continuous audits, as justified above. (Three decades of dataset annotations, our annotation tools, and annotation crowdsourcing mechanisms are all open sourced.) This can facilitate many new ecosystem-wide audits into data, beyond our work.
> > * Second, our license/terms annotations provide significant tooling to filter, search, and select thousands of datasets that match rigorous legal criteria. Very few datasets have been validated as free of legal/ethical restrictions. Especially for “ethical AI” startups, this is a real challenge. Our work takes considerable steps to provide this.
> > * Third, our dataset search tools help beyond legal criteria—practitioners can identify datasets with tasks/sources/languages they couldn’t find on HuggingFace (often because they weren’t labeled). Data creators can also trace all the datasets that used data from their website. These provide actionable insights to developers and creators.
> > * Fourth, our work demonstrates a critical need for more representative data, and illustrates exactly which languages/geographies have the widest gaps. For instance, our work shows that while Indonesian and Urdu are among the most spoken languages in the world, they are among the most relatively underrepresented in the language token count. This is also a call-to-action for the community to fill the most notable gaps uncovered by our ecosystem audit.
> > * Fifth, our work advocates for better documentation standards, especially for dataset restrictions, licenses, and terms. We provide a taxonomy of labels, instructions, and categories that we hope future work in this space will be able to build on.
> >
> > Finally, we’d like to thank you again for your helpful feedback. We believe these changes above will sharpen the work, and address your core recommendations. If our response addresses your key comments, we hope you will consider reflecting this in your score. Otherwise, we’d be happy to hear if you have further clarifications or suggestions.

---

> > > ### Author Response · Authors · 2024-11-25
> > >
> > > Dear Reviewer Kmty, we appreciate the time and effort you put into the review process. There are just under two days left. We would love to know if we have addressed your concerns, and if this has impacted your score. We've worked super hard on our paper updates, and appreciate your thoughts!

---

### Official Review · Reviewer_Yvyf · 2024-11-04

**Soundness:** 2
**Presentation:** 3
**Contribution:** 2
**Rating:** 5
**Confidence:** 2

**Summary:**

The paper conducts a large-scale dataset audit across text, speech, and video modalities, covering near 4000 public datasets between 1990-2024 by investigating their sources, use restrictions, and their geographical & linguistic representation. It is found that 1) Many multimodal ML datasets are from web-crawled and social media platforms; 2) Inconsistencies between dataset licenses and their source's restrictions prevail; and 3) Inequality in geographical representation remains very high, and geographical & linguistic representation has not significantly improved for many years.

**Strengths:**

• The paper is well-motivated: Most previous works has primarily focused on text datasets, or a single feature or dataset. The paper conducted a multimodal and multi-feature dataset, which addresses this gap.

• The scale of this study is impressive: It covers nearly 4000 datasets, 3 modalities (text, speech, video) and a time span of over 30 years (1990~2024), providing a comprehensive view to dataset provenance study.

• The study points out vulnerabilities of current dataset sources. For example, the paper discovers prevailing inconsistencies between dataset licenses and their source's restrictions, alarming the research community about potential legal breaches. The paper also reveals that many measures to address geographical & linguistic fairness have failed, surprisingly, which motivates the research community to retrospect these measures and potentially propose new (effective) ones.

**Weaknesses:**

• The paper mentions a rise in synthetic data, but does not give any in-depth analysis of synthetic data v.s. non-synthetic data. I think it is worthwhile to analyze the source distribution of synthetic data v.s. non-synthetic data, so that researchers can know the search space where they are more likely to acquire desired data when they need human-written or machine-generated data.

• The paper is not technically savvy, so it is crucial for the paper to highlight the importance of its insights. Given the large-scale of this study (nearly 4000 datasets) and the engagement of domain experts for annotation tasks, the study should be costly. But the paper did not make it very clear that why studying data provenance (probably with such high cost) is valuable at the first place. Why not use the same annotation labors to create new dataset resources instead, but studying the provenance of existing dataset (intuitively, the former one can be more valuable)?

**Questions:**

• Given that the study includes datasets from 67 countries, when studying data licenses or terms of use, are difference in legal regulations of copyrights between countries/regions taken into consideration?

---

> ### Author Response · Authors · 2024-11-21
>
> We would like to thank Reviewer Yvyf for their thoughtful and constructive feedback. We are particularly encouraged that they appreciated the scale of our analysis and found our results surprising and informative to the community’s future directions. We believe there are several feasible changes that will address their core concerns, which we outline here:
>
> **Comparing synthetic vs non-synthetic data—are they distributionally different?**
>
> We agree that this is an important point and discussing it would add useful insights for practitioners. We have added some key findings, taken from our audit, into the general response above. We are happy to flesh out these observations in a subsection in the paper, if the reviewer agrees that it addresses their recommendations.
>
>
> **Why allocate resources to study data provenance rather than create a new dataset?**
>
> This is a key motivating question for our work and one we passionately believe in. While we see the value in creating new datasets, we believe studying data provenance is critically undervalued. Within our work, we hope to make a convincing case for this. We propose making two changes to make this case: (a) adding the text below early in the paper, and (b) stressing in the discussion section the benefits/future work this analysis provides.
>
> “It is difficult to improve on what you cannot measure. Data is fundamental to machine learning systems, and understanding (measuring) their qualities in detail is critical to understanding many real concerns and risks: bias, representation, toxicity, legal risks, privacy, effects on data creators (copyright/licenses), contamination and of course, model capabilities. There is a long history of machine learning models being developed with such issues, which could have been diagnosed earlier (or prevented) had they leveraged training data audits [1, 2, 3, 4, 5]. These retrospective analyses have re-directed and guided subsequent training data curation practices in significant ways. Understanding data trends at the ecosystem level informs key gaps and limitations of the data that is out there, in ways that new dataset creation cannot. From this work, we empirically demonstrate key gaps in sourcing copyright/restriction-free data, and geographically representative data. Our released repository also provides practitioners with the tools to filter and search for the best available datasets that do fit these criteria—including those that are not as well known.”
>
> We believe our new contributions and tooling for analyzing data sourcing, restrictions, and representation meaningfully justify these resources. Please let us know if this is convincing, and addresses your concerns!
>
> **Given that the study includes datasets from 67 countries, when studying data licenses or terms of use, are differences in legal regulations of copyrights between countries/regions taken into consideration?**
>
> This is an important point as copyright laws vary across countries—we thank the reviewer for raising this. We note that the Berne convention complicates this analysis because signatories (which include a majority of nations) will give foreign works the same protection as domestic works. As a result, what matters is not only the origin of the data but also where the usage is taking place. We will clarify this complexity and cite some resources for practitioners to understand the law in some major jurisdictions: [6, 7, 8, 9, 10].

---

> > ### Author Response · Authors · 2024-11-21
> > **(Part 2—citations)**
> >
> > We’d like to thank the reviewer again for their constructive feedback and suggestions, which we believe will strengthen the work. If our response addresses your key comments, we hope you will consider reflecting this in your score. Otherwise, we’d be happy to hear if you have further clarifications or suggestions.
> >
> > 1. Abeba Birhane et al. "Multimodal datasets: misogyny, pornography, and malignant stereotypes." arXiv preprint arXiv:2110.01963 (2021).
> > 2. David, Emilia. "AI image training dataset found to include child sexual abuse imagery." The Verge. December 20, 2023. https://www.theverge.com/2023/12/20/24009418/generative-ai-image-laion-csam-google-stability-stanford.
> > 3. Joy Buolamwini, and Timnit Gebru. "Gender Shades: Intersectional Accuracy Disparities in Commercial Gender Classification." Proceedings of the 1st Conference on Fairness, Accountability and Transparency. Ed. Sorelle A. Friedler and Christo Wilson. Vol. 81, Proceedings of Machine Learning Research (2018): 77–91.
> > 4. Shayne Longpre, Robert Mahari, Anthony Chen, Naana Obeng-Marnu, Damien Sileo, William Brannon, Niklas Muennighoff, et al. "A Large-Scale Audit of Dataset Licensing and Attribution in AI." Nature Machine Intelligence 6, no. 8 (August 2024): 975–987. https://doi.org/10/gt8f5p
> > 5. Nishant Subramani, Sasha Luccioni, Jesse Dodge, and Margaret Mitchell. "Detecting Personal Information in Training Corpora: an Analysis." Proceedings of the 3rd Workshop on Trustworthy Natural Language Processing (TrustNLP 2023). Stroudsburg, PA, USA: Association for Computational Linguistics, 2023.
> > 6. Ginsburg, Jane C. "Copyright, common law, and sui generis protection of databases in the United States and abroad." U. Cin. L. Rev. 66 (1997): 151.
> > 7. Margoni, Thomas, and Martin Kretschmer. "A deeper look into the EU text and data mining exceptions: harmonisation, data ownership, and the future of technology." GRUR international 71.8 (2022): 685-701.
> > 8. Lin, Fen. "Digital intellectual property protection in China: Trends and damages in litigation involving the big five websites (2003–2013)." Asia Pacific Law Review 25.2 (2017): 149-169.
> > 9. Schirru, Luca, et al. "Text and Data Mining Exceptions in Latin America." IIC-International Review of Intellectual Property and Competition Law (2024): 1-30.
> > 10. Dermawan, Artha. "Text and data mining exceptions in the development of generative AI models: What the EU member states could learn from the Japanese “nonenjoyment” purposes?." The Journal of World Intellectual Property 27.1 (2024): 44-68.

---

> > > ### Author Response · Authors · 2024-11-25
> > >
> > > Dear Reviewer Yvyf, we appreciate the time and effort you put into the review process. There are just under two days left. We would love to know if we have addressed your concerns, and if this has impacted your score. We've worked super hard on our paper updates, and appreciate your thoughts!

---

### Author Response · Authors · 2024-11-21

We would like to thank the reviewers for their positive and constructive feedback! While we respond to each reviewer individually below, multiple asked about the synthetic data trends in our work. We agree this is an important topic, and would add useful insights for practitioners. Here we have consolidated some of the statistics uncovered by our audit:

* The top models used in generating datasets are mainly from OpenAI, with some open-weights models also popular. The top 5 consists of ChatGPT, version unspecified (15.0% of synthetic datasets), GPT-4 (14.4%), BART (10.1%), GPT-3 (8.3%) and GPT-3.5-Turbo (4.9%). 15.2% of datasets also used some kind of dataset-specific non-neural templating scheme.
* The average synthetic dataset also has notably longer turns (in tokens) than the average natural dataset: 1,756 tokens vs 1,065, a 64.8% increase.
* Synthetic dataset’s task distribution is shifted towards longer form, open-generation and creative tasks. For example, 88.1% of natural datasets contain classification tasks, vs only 66.3% of synthetic datasets. Natural data is also more likely to cover translation than synthetic data (72.4% of datasets vs only 22.9% of synthetic datasets). Both natural and synthetic data, however, had similar rates of creativity and Q&A tasks.
* Both natural and synthetic datasets are mostly English-only, though synthetic datasets are slightly more likely to be English-only than natural datasets (76.3% vs 69.6%).

All of these annotations, along with the analysis tools used to generate these findings will be open sourced. If the reviewers agree, we will add a subsection of the paper, dedicated to these empirical findings on the rise of synthetic data.

---

### Public Comment · ~Jonathan_Bennion1 · 2025-02-13
**Fallacy in the premise of this paper not addressed**

The premise of this paper focuses on data provenance, linguistic representation, and dataset transparency, without discussing the structural challenges of tokenization from a technical point of view.

Regarding the logical fallacy, calling an issue an "injustice" when it is fixable could be an example of a false dilemma, since assuming the problem is inherently unfair (rather than a technical or policy issue that could be addressed).

---

### Meta-Review · Area_Chair_uyu5 · 2024-12-20

**Metareview:**

The paper conducts a large-scale audit of nearly 4,000 public datasets (text, speech, and video) spanning the years 1990–2024 covering 608 languages, 798 data sources, 659 organizations, and 67 countries. It analyzes inconsistencies between dataset licenses and the terms of their data sources, noting that source terms are often more restrictive. It also reports that despite some increase in the number of languages and countries represented, the relative contribution remains Western-centric.

**Strengths identified**:
1. The paper is comprehensive in coverage.

2. It addresses a timely and critical topic relevant to the fast-paced growth of AI models and their reliance on datasets.

3. It provides novel insights into dataset sourcing, use restrictions, and geographical and linguistic representation.

4. The findings are valuable for advancing dataset governance and inclusivity and for understanding current and historical multimodal data conditions.

**Weakness that if addressed can strengthen the paper further**:
1. Limited depth in analyzing synthetic data, dataset tasks, and observed trends.

2. Lack of actionable outcomes or tools to address the issues raised: The paper lacks a clear call-to-action or concrete recommendations for improving the issues highlighted in the audit. It misses an opportunity to provide tools or frameworks that could automate similar audits in the future, making the findings more enduring.

3. Weak justification for the study's value compared to alternative efforts, like creating new datasets.

**Additional Comments On Reviewer Discussion:**

Two of the reviewers have not been able to revert to the responses (the authors have made efforts to provide detailed clarifications and also committed to future addition of discussions and analysis based on reviewer suggestions). The other two reviewers were satisfied.

---

### Decision · Program_Chairs · 2025-01-22

Accept (Poster)